# GraphSpa: Self-supervised Graph Sparsification for Robust Generalization

## Abstract

Graph sparsification has emerged as a promising approach to improve efficiency and remove redundant or noisy edges in large-scale graphs. However, existing methods often rely on task-specific labels, limiting their applicability in label-scarce scenarios, and they rarely address the residual noise that remains after sparsification. To address this issue, we aim to jointly consider both sparsity and robustness. In this work, we present GraphSpa, a self-supervised graph sparsification framework that constructs compact yet informative subgraphs without requiring labels, while explicitly mitigating residual noise. We formulate sparsification as a constrained optimization problem in which flatness is incorporated as part of the objective. Specifically, we address this problem by leveraging an augmented Lagrangian scheme to progressively satisfy the target sparsity. We also train the encoder to be robust to perturbations so that optimization is guided toward flatter regions of the loss landscape, reducing sensitivity to residual noise, and improving generalization. We theoretically demonstrate that this framework guarantees stable convergence while addressing both sparsity and robustness. Extensive experiments on benchmark datasets show that GraphSpa consistently outperforms baselines across various sparsity ratios and preserves cluster structures in t-SNE visualizations. Notably, it demonstrates strong and consistent performance on both large-scale and heterophilic datasets, validating its applicability in real-world scenarios. These results highlight GraphSpa as a principled and reliable framework for graph sparsification without labels and under residual noise.

## 1 Introduction

Graph Neural Networks (GNNs) have achieved remarkable success in a wide range of graph learning tasks, including node classification (Kipf & Welling, 2016), link prediction (Zhang & Chen, 2018), recommender systems (Ying et al., 2018), and social network modeling (Qiu et al., 2018). These advances demonstrate the strong potential of GNNs for analyzing complex relational data, yet scaling them to large real-world graphs remains challenging. As graph size increases, computational and memory costs grow rapidly, and real-world graphs often contain redundant or spurious edges (Li et al., 2024; Satuluri & Parthasarathy, 2011) that propagate misleading signals and degrade representation quality. Graph sparsification has emerged as a promising approach to mitigate these issues by removing redundant or noisy edges, thereby reducing overhead and yielding cleaner structural representations (Batson et al., 2013; Zheng et al., 2020). However, supervised sparsification methods rely on task-specific labels (Chen et al., 2021; Li et al., 2019), limiting their applicability in label-free scenarios such as recommender systems or social networks (Sobolevsky & Belyi, 2022; Guo et al., 2024). Meanwhile, unsupervised sparsification methods based on different structural properties have also been extensively explored, including path-based sparsification (Elkin & Neiman, 2017), topology-preserving sparsification (Meng et al., 2024; Loukas, 2019), and spectral sparsification based on effective resistance (Liu & Yu, 2022). However, these approaches rely on predefined structural properties and tailor sparsification toward specific notions of importance. As a result, they may preserve edges aligned with the chosen property while overlooking task-relevant information, and they do not explicitly address the residual noise that inevitably remains after sparsification.

Alongside reliance on labels, residual noise from sparsification poses another fundamental challenge. Since sparsification simplifies the graph structure, the diversity of propagation paths is reduced, making models more vulnerable to noisy edges (Dong & Kluger, 2023). With fewer effective

signals, over-parameterized GNNs tend to overfit and become more sensitive to residual noise (Zhou et al., 2018). This issue is particularly acute in domains such as social networks, where relationships themselves act as supervision signals. Spurious edges introduced by fake accounts or ephemeral connections distort the learning process and undermine downstream tasks such as community detection or node prediction (Wang et al., 2018). Addressing this challenge requires new sparsification methods that can effectively mitigate residual noise while operating without labels.

In this paper, we propose GRAPHSPA, a self-supervised graph sparsification framework designed to address both the reliance on labels and the vulnerability to residual noise. To overcome label dependence, GRAPHSPA explores diverse subgraph combinations and learns edge importance by comparing the mutual information shared between each sampled subgraph and the original graph. Each edge is modeled as a Bernoulli random variable, enabling probabilistic subgraph sampling, and the discrete edge-selection process is relaxed into continuous probabilities to allow gradient propagation. This formulation encourages broad exploration of structural variants early in training and gradually shifts focus toward meaningful structures, enabling the model to identify informative sparsified graphs without labels. Additionally, to mitigate residual noise after sparsification, GRAPHSPA applies Sharpness-Aware Minimization (SAM) (Foret et al., 2021) to the encoder, guiding optimization toward flatter minima through parameter perturbation. Although SAM is known for improving robustness to noise, sharpness-aware approaches remain underexplored in the context of sparsification. We formulate sparsification as a constrained optimization problem in which flatness is explicitly encouraged as an objective. To realize this formulation, GRAPHSPA adopts an augmented Lagrangian scheme (Boyd et al., 2011), enabling progressive rather than one-shot sparsity enforcement while guaranteeing convergence under both sparsity and robustness considerations.

We validate the effectiveness of GRAPHSPA through comprehensive experiments across a wide range of graph scenarios. On standard benchmark datasets such as Cora, Citeseer, and Pubmed, GRAPHSPA consistently outperforms existing baselines across different edge ratios while preserving meaningful structural information. Beyond these homophilous citation networks, GRAPHSPA also achieves strong performance on large-scale graphs such as Reddit and ogbn-arxiv as well as on heterophilic datasets, further demonstrating its scalability across diverse graph structures. In addition, robustness evaluations show that the learned representations remain stable even when various types of structural noise, including random noise, adversarial noise, and particularly homophily breaking noise, are injected after sparsification. These results indicate that GRAPHSPA is a robust and scalable sparsification framework suitable for label scarce graph learning settings.

Our main contributions are summarized as follows:

- We propose GRAPHSPA, a self-supervised graph sparsification framework that removes the reliance on labels and, to the best of our knowledge, is the first to explicitly address the harmful effect of residual noisy edges that remain after sparsification.

- We propose a sparsification framework that unifies augmented Lagrangian based constrained optimization with flatness-aware training, achieving both sparsity and robustness under provable convergence guarantees.

- We conduct extensive experiments on multiple benchmarks, demonstrating that GRAPHSPA consistently outperforms baselines across edge ratios, preserves structural integrity, and achieves strong generalization under noisy conditions.

## 2 RELATED WORKS

**Graph Self-Supervised Learning (Graph SSL)** has emerged as a powerful paradigm in graph neural network (GNN) research, attracting significant attention from both academia and industry. In graph SSL, the model is trained through well-designed auxiliary tasks, where supervisory signals are automatically generated from the data without requiring manual labels (Li et al., 2022b; Liu et al., 2021). Among various approaches, contrastive learning has proven to be one of the most successful strategies for graph data (Velickovic et al., 2019; Xu et al., 2021; Zeng & Xie, 2021). Its key idea is to maximize the similarity between representations of two different augmented views of the same graph, typically by maximizing their mutual information (van den Oord et al., 2018). Such methods have achieved state-of-the-art performance in diverse graph-based downstream tasks, but research that combines graph SSL with graph sparsification remains relatively limited.

**Graph Sparsification** aims to construct a sparser graph by removing a subset of edges from the original graph. This reduces storage cost, accelerates GNN training and inference, and alleviates the impact of redundant or noisy edges. However, many existing sparsification methods rely heavily on sufficient label information, which is often scarce in real-world scenarios such as recommender systems or social networks (Yang et al., 2016; Hu et al., 2020). In label-scarce settings, sparsification is typically performed by preserving certain structural properties of the graph, such as degree distribution, local topology, path distance, or spectral characteristics. However, the edges that are important for downstream tasks vary significantly across applications, and sparsifying the graph based on a single structural property cannot capture such diversity. As a result, important edges may be removed or irrelevant ones retained, limiting the generality of property-preserving sparsifiers. Furthermore, existing studies have shown that GNNs are highly vulnerable to structural noise (Li et al., 2022a) or intentionally injected adversarial perturbations (Chen et al., 2020). Remaining noisy edges can even distort node representations and severely degrade the generalization performance of GNNs (Zügner et al., 2018). Prior work also reports that conventional sparsification methods often fail to consistently eliminate such harmful edges (Chen et al., 2021). Therefore, to obtain sparsified graphs that remain reliable in realistic environments, it is essential to develop a new sparsification strategy that does not depend on predefined structural properties while effectively reducing the model's sensitivity to remaining noisy edges in practice, particularly under challenging conditions.

## 3 PRELIMINARIES

To ground our method, we first formalize the problem of graph sparsification and review the principle of flatness-aware optimization. These preliminaries establish the foundation for GRAPHSPA, which integrates self-supervised sparsification with flatness-aware training to address residual noise.

### 3.1 PROBLEM SETUP

We begin by representing an undirected input graph $G = (V, E)$, where $V$ is the set of $N$ vertices and $E$ is the set of edges. The graph structure is described by the adjacency matrix $A \in \mathbb{R}^{N \times N}$, where $A[i, j] = 1$ if $(i, j) \in E$ and 0 otherwise. Each vertex $v \in V$ is associated with a feature vector $\mathbf{x}_v \in \mathbb{R}^F$, and the feature matrix is denoted as $X \in \mathbb{R}^{N \times F}$.

Given $(A, X)$, GNN $f_\theta$ learns node representations by iteratively aggregating information from neighbors across layers. At the $l$-th layer, the representation of node $v$ is updated as:

$$h_v^{(l+1)} = \psi\Big(h_v^{(l)}, \phi\{h_u^{(l)} \mid u \in N_v\}\Big), \tag{1}$$

where $\phi$ denotes an aggregation function over neighbors, $\psi$ combines the previous representation of $v$ with the aggregated messages, and $h_v^{(0)} = \mathbf{x}_v$ is the initial representation.

The goal of graph sparsification is to learn a function

$$\mathcal{P} : G \to G_s, \tag{2}$$

where $G_s \subseteq G$ is a sparsified subgraph that preserves as much informative structure of $G$ as possible. Formally, $G_s = (V, E_s)$ is defined by an adjacency matrix $A_s \in \{0, 1\}^{|E|}$, where $A_s[i, j] = 1$ if the edge $(i, j) \in E_s$ is kept and 0 otherwise. An edge retention ratio $r \in (0, 1)$ controls the proportion of edges retained, and $G_s$ keeps $r\%$ of the original edges. In the self-supervised setting, no label information such as node labels is available. Instead, the sparsification mechanism has to identify and retain informative edges without supervision.

### 3.2 SHARPNESS-AWARE MINIMIZATION

**Sharpness-Aware Minimization (SAM)** aims to find loss minima that are not only high-performing but also insensitive to parameter perturbations, thereby improving generalization and robustness (Foret et al., 2021). Formally, SAM solves the following min–max optimization problem:

$$\min_{\theta} \max_{\|\epsilon\| \leq \rho} \mathcal{L}(\theta + \epsilon), \tag{3}$$

where $\mathcal{L}(\theta)$ is the training loss for parameters $\theta$, and $\epsilon$ denotes parameter perturbations within an $\ell_p$ ball of radius $\rho$, which determines the maximum perturbation size. The inner maximization seeks the

worst-case performance under perturbations, while the outer minimization finds parameters robust to such perturbations. To efficiently approximate the inner maximization, SAM uses a first-order Taylor expansion. The perturbation that maximally increases the loss is estimated as:

$$\hat{\epsilon} = \rho \cdot \frac{\nabla_\theta \mathcal{L}(\theta)}{\|\nabla_\theta \mathcal{L}(\theta)\|_2} \approx \arg \max_{\|\epsilon\|_p \leq \rho} \mathcal{L}(\theta + \epsilon), \tag{4}$$

At training step $t$, SAM is implemented via the following iterative process:

$$\epsilon_t = \nabla_\theta \mathcal{L}(\theta_t), \quad \hat{\epsilon}_t = \rho \cdot \frac{\epsilon_t}{\|\epsilon_t\|_2}, \quad \omega_t = \nabla_\theta \mathcal{L}(\theta_t + \hat{\epsilon}_t), \quad \theta_{t+1} = \theta_t - \eta \cdot \omega_t, \tag{5}$$

where $\epsilon_t$ is the perturbation gradient, $\hat{\epsilon}_t$ is the normalized perturbation within the $\rho$-ball, $\omega_t$ is the updating gradient evaluated at the perturbed parameters, and $\eta$ is the learning rate. By updating parameters using gradients computed at perturbed weights, SAM explicitly encourages convergence to flat minima, where the loss landscape varies smoothly under small perturbations, thereby improving generalization and robustness across diverse domains (Foret et al., 2021; Baek et al., 2024).

## 4 GRAPHSPA

In this section, we introduce GRAPHSPA, a self-supervised graph sparsification framework that explicitly addresses residual noise while preserving the structural information of the original graph. GRAPHSPA formulates sparsification with a target edge budget as a constrained optimization problem. Instead of relying on labels, each edge is modeled as a differentiable Bernoulli random variable, and the loss is defined as the mutual information between the sampled subgraph and the original graph. By maximizing this objective, the framework learns edge importance scores and identifies particularly informative structures. Based on these importance scores, we adopt an augmented Lagrangian approach with convergence guarantees to gradually impose sparsity during optimization, rather than removing edges in a one-shot manner. Moreover, GRAPHSPA further integrates flatness-aware training into the sparsification process to optimize the encoder in a way that effectively reduces sensitivity to residual noise, thereby ensuring robust generalization even without labels.

### 4.1 PROBLEM FORMULATION

**Self-Supervised Objective.** We adopt a self-supervised strategy to preserve the essential information of the original graph $G$. Specifically, we maximize the mutual information between the original graph $G$ and the sparsified graph $G_s$ by adopting the InfoNCE loss (van den Oord et al., 2018).

Let node embeddings be $\mathbf{H} = f_\theta(X, A_s)$ obtained from a GNN encoder parameterized by $\theta$, where $h_v$ denotes the embedding of node $v \in \mathcal{V}$. The pair $(G, G_s)$ is treated as a positive sample, while negative samples $\tilde{G}_s$ are generated by randomly dropping a portion of edges from $G$. The contrastive loss is then defined as

$$\mathcal{L} = -\sum_{v \in \mathcal{V}} \log \frac{\exp(\text{sim}(h_v^G, h_v^{G_s})/\beta)}{\sum_{u \in \mathcal{V}} \exp(\text{sim}(h_v^G, h_u^{G_s})/\beta)}, \tag{6}$$

where $\text{sim}(\cdot, \cdot)$ is a similarity function such as cosine similarity and $\beta$ is a temperature parameter. This loss encourages the embeddings from $G_s$ to remain consistent with those from $G$, ensuring that sparsification retains informative edges without using labels.

**Edge Importance Learning via Bernoulli Subgraph Sampling.** At each training iteration, we need to construct a sparsified subgraph to learn importance of individual egdes. A naive approach would be to randomly sample edges from the original graph, which incurs an exponential search space of $2^{|E|}$ possible subgraphs and does not allow gradient propagation since edge selection is a discrete 0-1 decision. To address this, we relax the binary mask into a continuous probability through a learnable logit $x_{ij}$, which reflects the latent importance of edge $(i, j)$. Through the Gumbel-Softmax relaxation (Jang et al., 2017), we obtain a continuous importance score $s_{ij} \in (0, 1)$:

$$s_{ij} = \sigma\left(\frac{\log \xi_{ij} - \log(1 - \xi_{ij}) + x_{ij}}{\tau}\right), \quad \xi_{ij} \sim \mathcal{U}(0, 1), \tag{7}$$

where $\sigma(\cdot)$ is the sigmoid function and $\tau > 0$ is a temperature parameter. We initialize $x_{ij} = 0$ so that all edges start with equal importance.

The importance score $s_{ij}$ serves a dual role. It provides a differentiable relaxation of binary edge selection, and it determines the probability that edge $(i, j)$ is selected when constructing a subgraph. Formally, each edge is sampled according to a Bernoulli distribution with selection probability $s_{ij}$:

$$A_s(i, j) \sim \text{Bernoulli}(s_{ij}), \quad \forall (i, j) \in \mathcal{E}. \tag{8}$$

In other words, edge $(i, j)$ is included in the sampled subgraph with probability $s_{ij}$ and excluded otherwise. By interpreting $s_{ij}$ as both a trainable relaxation and a sampling probability, the model can generate subgraphs in a stochastic manner. This sampling mechanism enables exploration of diverse structural variants, ensuring that even edges with low scores are occasionally selected. As perfectly identifying and removing noisy edges is infeasible, this strategy prevents the model from prematurely discarding potentially informative connections while still encouraging sparsification.

In practice, we start from a high temperature $\tau$ to encourage exploration of diverse subgraphs and gradually decrease it following a cosine scheduling strategy. This allows the model to explore structural variants more freely in the early stage of training, while focusing on more deterministic edge selection in the later stage as the sparsity constraint becomes progressively tighter. Details of the ablation study on the temperature scheduling strategy are provided in Appendix D.5.

**Flatness-Aware Training.** To enhance robustness against residual noise and improve generalization performance, we adopt a flatness-aware training strategy based on a min–max optimization. Specifically, the sparsified subgraph $G_s$ is sampled from the original graph $G$ according to the importance score $s_{ij}$. We then optimize the following objective:

$$\min_{\theta} \max_{\|\epsilon\|_p \leq \rho} \mathcal{L}(G_s, \theta + \epsilon), \tag{9}$$

where $\theta$ denotes the encoder parameters, $\epsilon$ is a perturbation vector, and $\rho$ is the perturbation radius. The inner maximization corresponds to injecting perturbations into the encoder parameters, which simulates worst-case deviations during training and mimics the corrupted message passing caused by noisy edges commonly observed in practical scenarios. The outer minimization then drives the model to learn representations that remain stable under such perturbations, thereby improving generalization performance and reducing sensitivity to residual noise. In other words, the encoder is guided toward flat minima that generalize well under residual noise conditions during training.

## 4.2 Constrained Optimization

To mitigate the irreversible information loss caused by one-shot criterion-based sparsification, our key idea is to gradually impose substantial sparsity onto the edges while maximally preserving information during training through a simple iterative process designed for stability. However, the restriction on the number of edges is inherently non-differentiable due to the discrete nature of the $\ell_0$ constraint, which makes direct optimization infeasible. A standard approach for such constrained problems is to employ Lagrangian duality or projected gradient descent. Yet, the discrete nature of the $\ell_0$-norm makes Lagrangian duality infeasible, while projected gradient descent, despite its efficiency, often struggles with highly non-convex objectives in neural network optimization.

To balance the smooth optimization of Lagrangian methods with the efficiency of projection, we adopt an augmented Lagrangian relaxation inspired by ADMM (Boyd et al., 2011). To impose sparsity, we introduce an auxiliary variable $z$ with the equality constraint $x = z$, where $z$ periodically stores the projected sparse solution. This leads to the following problem, where the sparsity constraint $\|z\|_0 \leq r|E|$ ensures that only $r \times |E|$ edges are retained:

$$\min_{x,z} \max_{\|\epsilon\|_p \leq \rho} \mathcal{L}(G_s, \theta + \epsilon) + I_{\|z\|_0 \leq r|E|}(z), \quad \text{s.t. } x = z, \tag{10}$$

where $I_{\|z\|_0 \leq r|E|}(z)$ is the indicator function of the sparsity constraint:

$$I_{\|z\|_0 \leq r|E|}(z) := \begin{cases} 0, & \|z\|_0 \leq r|E|, \\ \infty, & \text{otherwise.} \end{cases} \tag{11}$$

---

**Algorithm 1** GRAPHSPA

---

**Require:** Target edge ratio $r$, total iterations $T$, dual-update interval $K$, penalty parameter $\lambda$, perturbation radius $\rho$, temperature $\tau$
1: **Initialize** $x^{(0)}$
2: $u = 0$
3: **for** $t = 0$ in $T - 1$ **do**
4:     **for** each edge $(i, j) \in E$ **do**
5:         $\xi_{ij} \sim \mathcal{U}(0, 1)$
6:         $s_{ij}^{(t)} \leftarrow \sigma\left( \dfrac{\log \xi_{ij} - \log(1 - \xi_{ij}) + x_{ij}^{(t)}}{\tau} \right)$
7:         $A_s^{(t)}(i, j) \sim \text{Bernoulli}(s_{ij}^{(t)})$
8:     **end for**
9:     Construct subgraph $G_s^{(t)} = (V, A_s^{(t)})$
10:     **if** $t \bmod K = 0$ **then**
11:         $z^{(t+1)} \leftarrow \text{Proj}_{\|z\|_0 \leq r|E|}\left(x^{(t)} + u^{(t)}\right)$
12:         $u^{(t+1)} \leftarrow u^{(t)} + x^{(t)} - z^{(t+1)}$
13:     **else**
14:         $z^{(t+1)} \leftarrow z^{(t)}, \quad u^{(t+1)} \leftarrow u^{(t)}$
15:     **end if**
16:     $x^{(t+1)} \leftarrow x^{(t)} - \eta^{(t)}\left( \nabla_x \mathcal{L}(G_s^{(t)}, \theta^{(t)}) + \lambda(x^{(t)} - z^{(t)} + u^{(t)}) \right)$
17:     $\hat{\epsilon} \leftarrow \rho \cdot \dfrac{\nabla_\theta \mathcal{L}(G_s^{(t)}, \theta^{(t)})}{\|\nabla_\theta \mathcal{L}(G_s^{(t)}, \theta^{(t)})\|_2}$
18:     $\theta^{(t+1)} \leftarrow \theta^{(t)} - \eta^{(t)} \nabla_\theta \mathcal{L}(G_s^{(t)}, \theta^{(t)} + \hat{\epsilon})$
19: **end for**
20: **return** $\text{Proj}_{\|z\|_0 \leq r|E|}\left(x^{(T)}\right)$

---

To enforce $x = z$ during optimization, we introduce a scaled dual variable $u$ and add a quadratic penalty term $\frac{\lambda}{2}\|x - z\|_2^2$, yielding the augmented Lagrangian relaxation:

$$\max_u, \min_{x,z} \left( \mathcal{L}(x, z, u) := \max_{\|\epsilon\|_p \leq \rho} \mathcal{L}(G_s, \theta + \epsilon) + I_{\|z\|_0 \leq r|E|}(z) - \frac{\lambda}{2}\|u\|_2^2 + \frac{\lambda}{2}\|x - z + u\|_2^2 \right). \quad (12)$$

Applying alternating minimization with respect to $x$ and $z$, and dual ascent on $u$, we obtain the following optimization subproblems:

$$x_{k+1}, z_{k+1} = \arg\min_{x,z} \max_{\|\epsilon\|_p \leq \rho} \left( \mathcal{L}(G_s, \theta + \epsilon) + I_{\|z\|_0 \leq r|E|}(z) + \frac{\lambda}{2}\|x - z + u_k\|_2^2 \right),$$
$$u_{k+1} = \arg\max_u \frac{\lambda}{2}\|x_{k+1} - z_{k+1} + u\|_2^2 - \frac{\lambda}{2}\|u\|_2^2. \quad (13)$$

The $z$-update corresponds to a projection due to the indicator function, and the $u$-update reduces to a simple dual ascent step. Therefore, the iterative scheme becomes:

$$x_{k+1} = \arg\min_x \max_{\|\epsilon\|_p \leq \rho} \left( \mathcal{L}(G_s, \theta + \epsilon) + \frac{\lambda}{2}\|x - z_k + u_k\|_2^2 \right),$$
$$z_{k+1} = \text{Proj}_{\|z\|_0 \leq r|E|}(x_{k+1} + u_k), \quad (14)$$
$$u_{k+1} = u_k + x_{k+1} - z_{k+1}.$$

Since the $x$-minimization cannot be solved in closed form, we approximate it by a single gradient descent step on the objective. This yields the practical update rules:

$$x_{k+1} = x_k - \eta\left( \nabla_x \mathcal{L}(G_s, \theta_k) + \lambda(x_k - z_k + u_k) \right),$$
$$z_{k+1} = \text{Proj}_{\|z\|_0 \leq r|E|}(x_{k+1} + u_k), \quad (15)$$
$$u_{k+1} = u_k + x_{k+1} - z_{k+1}.$$

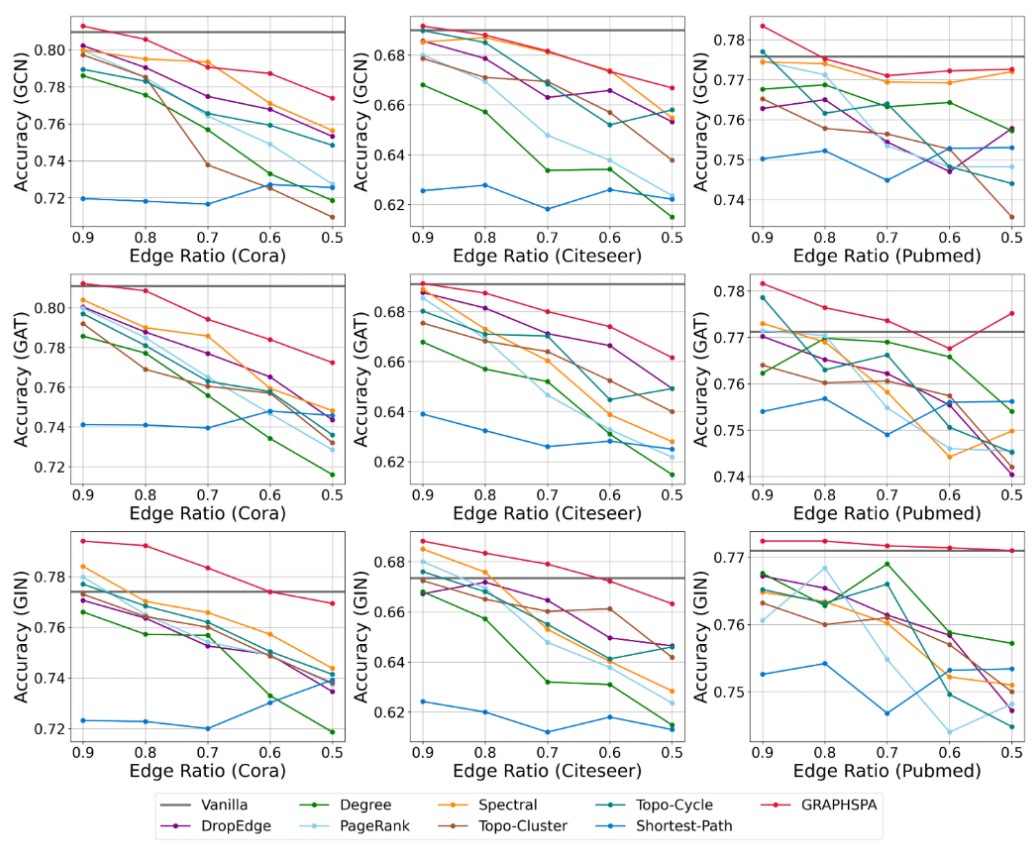

Figure 1: Node classification accuracy under different graph sparsity ratios on GCN/GAT/GIN across the Cora/Citeseer/Pubmed datasets. Results are reported as mean over five random seeds, and statistical significance is validated with paired t-tests ($p < 0.05$).

During training, this procedure gradually aligns $x$ with $z$, allowing continuous optimization between projection steps, and avoids the irreversible information loss of one-shot sparsification. In this way, the framework achieves progressive sparsification that preserves essential structural information under a hard $\ell_0$ constraint, while benefiting from the stability of augmented Lagrangian optimization.

### 4.3 NOISE-RESILIENT ENCODER OPTIMIZATION

While updating $(x, z, u)$ with the augmented Lagrangian scheme, we simultaneously update the model parameters $\theta$ using the same loss function $\mathcal{L}$ applied to the sparsified subgraph $G_s$. By injecting perturbations into the GNN parameters, the encoder is trained in a flatness-aware manner, which reduces its sensitivity to residual noisy edges. As a result, the learned representations become more robust and generalizable, achieving improved performance even under conditions where residual noise persists in the graph. The perturbation vector is approximated as:

$$\hat{\epsilon} = \rho \frac{\nabla_\theta \mathcal{L}(G_s, \theta_k)}{\|\nabla_\theta \mathcal{L}(G_s, \theta_k)\|_2}, \tag{16}$$

and the parameter update is given by

$$\theta_{k+1} = \theta_k - \eta \nabla_\theta \mathcal{L}(G_s, \theta_k + \hat{\epsilon}). \tag{17}$$

***Intuition.*** Each iteration of GRAPHSPA proceeds as follows: (i) a subgraph $G_s$ is sampled using the current edge probabilities from $x$, with a high initial temperature gradually annealed via cosine scheduling to balance exploration and exploitation, (ii) the auxiliary variables $z$ and $u$ are updated

every $K$ steps to enforce the hard $\ell_0$ constraint through projection and dual ascent, (iii) the edge logits $x$ are updated while staying close to the sparsity-projected proxy $z$ and simultaneously maximizing mutual information with the original graph to preserve informative structures, and (iv) the model parameters $\theta$ are optimized toward flatter minima via perturbation-based updates, reducing sensitivity to residual noisy edges and mitigating overfitting. The overall procedure of our framework is summarized in Algorithm 1. We provide a theoretical guarantee that the $x$-minimization converges during training. The detailed proof of convergence is deferred to Appendix A.

## 5 EXPERIMENTS

In this section, we present experiments to validate the effectiveness of the proposed framework. We first introduce the experimental settings, then compare our method with several baselines that do not use labels, and finally provide analysis to highlight its advantages in terms of performance, generalization, and applicability under noisy graph settings.

**Datasets & Models.** We evaluate our framework on three transductive benchmark datasets: Cora, Citeseer, and Pubmed (Kipf & Welling, 2016). To examine scalability on large-scale graphs, we additionally evaluate on Reddit (Hamilton et al., 2017), which follows an inductive setting, and ogbn-arxiv (Hu et al., 2020). We adopt the public splits for all datasets, and the dataset statistics are summarized in Table 2. For backbone models, we use Graph Convolutional Network (GCN) (Kipf & Welling, 2016), Graph Attention Network (GAT) (Veličković et al., 2018), and Graph Isomorphism Network (GIN) (Xu et al., 2019). For the larger datasets, we follow standard practice and use GraphSAGE (Hamilton et al., 2017) as the backbone model.

**Baselines.** We compare our method against representative sparsification strategies. Vanilla uses the original graph without modifying edges and serves as a reference for evaluating the effect of sparsification. DropEdge (Rong et al., 2020) removes edges uniformly at random to reduce overall edge density. Topology-preserving sparsification includes degree-based methods (Batagelj & Zaversnik, 2003) and techniques designed to maintain important local structural organization during sparsification. This category also includes Topo-Cycle (Loukas, 2019) and Topo-Cluster (Meng et al., 2024), which aim to preserve characteristic neighborhood patterns and meaningful topological structures. Spectral sparsification is represented by effective-resistance (ER) based approaches (Liu & Yu, 2022), where edge importance is computed using analogies from electrical networks to preserve the Laplacian quadratic form. Path-based sparsification includes Shortest-Path spanner constructions (Elkin & Neiman, 2017), which preserve approximate pairwise distances under bounded stretch constraints, as well as PageRank-based sparsification (Page et al., 1999), which favors edges associated with structurally influential nodes based on stationary random-walk probabilities. To ensure consistent sparsity levels across all baselines, we lightly modify the algorithms that do not originally support explicit sparsity control so that they can produce graphs that match the target sparsity ratio. Further implementation details are provided in Appendix B.1.

### 5.1 PERFORMANCE ANALYSIS

Figure 1 reports the node classification accuracy under different edge retention ratios $r$, where $r$ denotes the proportion of edges retained after sparsification. Overall, our method demonstrates consistently strong performance across all sparsity ratios and datasets. MI objective maximizes the shared information between the original and sparsified graphs, encouraging the model to preserve signals such as node features, local connectivity patterns, multi-hop dependencies, embedding geometry, and broader semanctic or structural information. Combined with the augmented Lagrangian–based constrained optimization, our approach progressively satisfies the target sparsity while reliably retaining high-importance edges, thereby maintaining robust performance even as sparsity increases.

In contrast, traditional sparsification methods focus on preserving structural properties such as Shortest-Path distances, spectral characteristics, or local topological patterns. Since the importance of these properties varies across graph types, such methods often fail to operate consistently in real-world settings where multiple structural patterns coexist. Our MI-driven formulation avoids making such assumptions and instead preserves the information that is inherently important to the learned representations. At a light sparsification level of $r = 0.9$, our method not only mitigates the negative impact of edge removal but also consistently outperforms the vanilla models across all three

| Method | ogbn-arxiv (Accuracy ↑) | | | | | Reddit (Accuracy ↑) | | | | |
|---|---|---|---|---|---|---|---|---|---|---|
| Edge Ratio | 0.9 | 0.8 | 0.7 | 0.6 | 0.5 | 0.9 | 0.8 | 0.7 | 0.6 | 0.5 |
| DropEdge | 68.89 ↓ 2.36 | 69.20 ↓ 2.05 | 68.52 ↓ 2.73 | 67.68 ↓ 3.57 | 66.42 ↓ 4.83 | 93.60 ↓ 1.81 | 93.89 ↓ 1.52 | 93.47 ↓ 1.94 | 93.22 ↓ 2.19 | 92.78 ↓ 2.63 |
| Degree | 70.56 ↓ 0.69 | 70.01 ↓ 1.24 | 69.44 ↓ 1.81 | 68.39 ↓ 2.86 | 67.23 ↓ 4.02 | 94.33 ↓ 1.08 | 94.11 ↓ 1.30 | 93.94 ↓ 1.47 | 93.69 ↓ 1.72 | 93.21 ↓ 2.20 |
| PageRank | 69.48 ↓ 1.77 | 68.94 ↓ 2.31 | 68.20 ↓ 3.05 | 67.31 ↓ 3.94 | 66.18 ↓ 5.07 | 93.60 ↓ 1.81 | 93.41 ↓ 2.00 | 93.20 ↓ 2.21 | 92.98 ↓ 2.43 | 92.51 ↓ 2.90 |
| GRAPHSPA | **72.82** ↑ 1.57 | **72.15** ↑ 0.90 | **71.57** ↑ 0.32 | **70.84** ↓ 0.41 | **69.64** ↓ 1.61 | **96.18** ↑ 0.77 | **95.97** ↑ 0.56 | **95.89** ↑ 0.48 | **95.58** ↑ 0.17 | **95.24** ↓ 0.17 |
| Vanilla | 71.25 ± 0.08 | | | | | 95.41 ± 0.05 | | | | |

Table 1: Node classification performance on ogbn-arxiv and Reddit under different edge sparsity ratios. All methods are trained using GraphSAGE, and the reported results denote the mean accuracy over five random seeds. Out-of-time methods are excluded from the comparison.

datasets. This indicates that removing redundant or noisy edges through sparsification enables the backbone to learn cleaner and more informative representations. When the edge ratio is reduced to $r = 0.5$, Cora and Citeseer exhibit performance degradation, which is expected since graphs with substantially fewer edges are more likely to lose essential structural information. Nevertheless, our method shows a much slower decline compared to existing sparsification approaches that remove edges based on specific structural properties, resulting in more stable performance across diverse conditions. Finally, on the Pubmed dataset, which contains far more edges than Cora or Citeseer, our method achieves performance comparable to the vanilla backbone even at $r = 0.5$. This suggests that the advantages of our method become increasingly pronounced as graph size grows.

Table 1 demonstrates that our method consistently maintains strong performance even on large-scale graphs such as ogbn-arxiv and Reddit. In contrast, several existing sparsification techniques were unable to complete within the time budget due to their rapidly increasing computational requirements on large graphs, and thus could not be included in the final comparison. For example, shortest-path spanners require computations close to all-pairs shortest paths, and ER-based spectral sparsifiers incur substantial memory and computational overhead when estimating effective resistance. These characteristics make many structure-preserving sparsifiers impractical in large-scale settings. Overall, the results indicate that our method achieves both efficiency and high performance in large graph scenarios, highlighting its practical applicability to real-world, large-scale networks.

## 5.2 ROBUSTNESS TO NOISY EDGES

Existing studies have shown that GNNs are not robust to structural noise (Li et al., 2022a) or intentionally injected adversarial perturbations (Chen et al., 2020). Such noise can distort node representations and significantly degrade the generalization performance of GNNs (Zügner et al., 2018). Furthermore, prior work has reported that conventional sparsification methods often fail to consistently remove harmful edges (Chen et al., 2021), highlighting the need for sparsification techniques that make the resulting model less sensitive to remaining noisy connections.

To evaluate the robustness of our approach under noisy conditions, we first sparsify the original graph by retaining $r = 0.7$ of the edges and then inject three types of structural noise. Random noise is generated by inserting spurious edges between randomly selected node pairs following the protocol of (Jin et al., 2021). Adversarial noise is introduced by perturbing the graph structure in a way that intentionally misleads the classifier, based on the Metattack framework (Zügner & Günnemann, 2019). Homophily-breaking noise is produced by adding edges that connect nodes with dissimilar labels or weak semantic similarity, thereby disrupting local structural consistency, as discussed in (Bo et al., 2021). The noise ratio $r_{noise} \in \{0.1, 0.2, 0.3, 0.4, 0.5\}$ denotes the proportion of injected edges relative to the number of edges remaining after sparsification. Using a GCN model, we report the average classification accuracy over five runs on the Pubmed dataset.

Figure 2 presents the results. The experimental results show that existing sparsification methods, which are designed to preserve specific structural properties, are highly vulnerable when injected noise disrupts the very properties they aim to maintain. Both homophily-breaking and adversarial perturbations distort local structural consistency and induce misleading message-passing patterns, leading to substantial performance degradation for property-based sparsifiers. In contrast, GRAPHSPA does not rely on preserving any predefined structural property. Instead, it performs sparsification by maximizing the shared information between the original and sparsified graphs, encouraging the retention of rich, representation-level signals learned by the encoder rather than

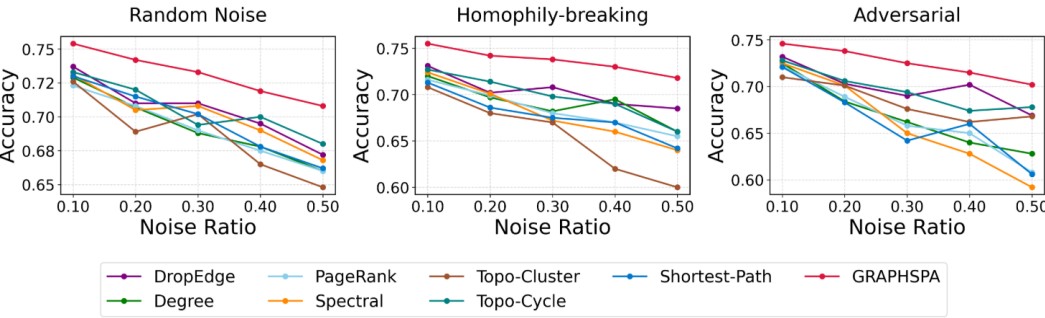

Figure 2: Node classification accuracy of GCNs on the Pubmed dataset under three types of injected structural noise after 30% edge sparsification. Results are reported as mean accuracy over five random seeds, with statistical significance assessed using $p < 0.05$.

enforcing superficial structural similarity. These signals include node features, local connectivity patterns, multi-hop dependencies, and geometric or semantic relationships within the embedding space, enabling a more comprehensive form of information preservation. Furthermore, GRAPHSPA adopts a flatness-aware optimization during sparsification, which guides the encoder toward flatter and more stable minima. This joint sparsification–stabilization process makes the learned representations more resilient to noisy edges, preventing the encoder from overfitting to spurious structures and maintaining robust performance even when substantial structural noise is present. Consequently, although accuracy gradually decreases as the noise ratio increases, the decline remains consistently smaller than that of all baseline methods, and GRAPHSPA achieves the highest accuracy across all three noise types. These results demonstrate that the proposed approach produces sparsified graphs that remain reliable in realistic settings where diverse forms of structural noise naturally arise.

### 5.3 QUALITATIVE ANALYSIS

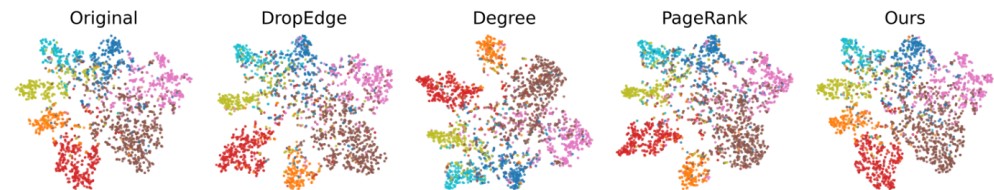

Figure 3: t-SNE visualization of node embeddings on the Pubmed after 50% edge sparsification.

Figure 3 presents the 2D t-SNE projections of node embeddings after removing 50% of the edges using different methods. As observed, the embeddings from our method exhibit a clustering structure consistent with the original graph, whereas other baselines show altered cluster distributions. This indirectly demonstrates that our sparsification strategy can more effectively preserve the structural information of the original graph. Moreover, our method produces compact yet informative subgraphs, enabling reliable graph learning without labels even under noisy conditions.

## 6 CONCLUSION

In this work, we presented GRAPHSPA, a self-supervised framework that effectively tackles the dual challenges of label scarcity and residual noise. We formulated the sparsification process as a constrained optimization problem using an augmented Lagrangian scheme to progressively learn compact structures and achieve target sparsity. Concurrently, we integrated flatness-aware training to resist parameter perturbations, explicitly mitigating the impact of residual noise on generalization. Crucially, we theoretically demonstrated that this joint optimization framework guarantees stable convergence while simultaneously balancing sparsity and robustness. Extensive experiments on large-scale and heterophilic datasets validate GRAPHSPA's superior efficiency and structural preservation, establishing it as a principled and reliable solution for real-world graph learning.

## ACKNOWLEDGMENT

### LLM USAGE

Large Language Models (LLMs) were used strictly as general-purpose assistants for writing refinement, retrieval of related work, and high-level research ideation. All technical contributions, experimental designs, and analyses were developed and validated exclusively by the authors. The LLM did not generate original scientific content, nor did it influence the methodological or empirical decisions of the work. The authors take full responsibility for all content presented in this paper.

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

SUPPLEMENTARY MATERIALS

# A    CONVERGENCE ANALYSIS

In this section, we establish the convergence of our flatness-aware sparsification framework. Our proof builds on the augmented Lagrangian method (Boyd et al., 2011) and extends the fundamental convergence analysis of sharpness-aware minimization (Khanh et al., 2024), thereby providing theoretical guarantees for the stability of our approach.

## A.1    ASSUMPTIONS

**Assumption A.1.** (Smoothness and Weak Convexity) The loss $\mathcal{L}(G_s^{(t)}, \theta)$ is $\beta$-smooth and $\mu$-weakly convex in $x$.

**Assumption A.2.** (Lipschitz Gradient) The gradient of $\mathcal{L}$ with respect to $x$ is Lipschitz, and stochastic gradients (if any) are unbiased and have bounded variance.

**Assumption A.3.** (Step Size) The step size $\{\eta^{(t)}\}$ is diminishing, satisfying

$$\sum_{t=1}^{\infty} \eta^{(t)} = \infty, \qquad \sum_{t=1}^{\infty} (\eta^{(t)})^2 < \infty.$$

**Assumption A.4.** (Perturbation Radius) The perturbation radius $\{\rho^{(t)}\}$ applies only to $\theta$-updates and is bounded and/or diminishing, satisfying

$$\limsup_{t \to \infty} \rho^{(t)} < \tfrac{1}{\beta}, \qquad \sum_{t=1}^{\infty} \eta^{(t)} \rho^{(t)} < \infty.$$

**Assumption A.5.** (Strong Convexity of the Augmented Term) The penalty parameter satisfies $\lambda > \mu$, ensuring a strong convexity component in the augmented Lagrangian.

## A.2    SMOOTHNESS AND CONVEXITY OF THE AUGMENTED LAGRANGIAN

**Lemma A.1.** Under Assumptions A.1–A.5, the augmented Lagrangian
$$\hat{\mathcal{L}}(x, z, u, \theta) = \mathcal{L}(G_s^{(t)}, \theta) + I_{\|z\|_0 \le r|E|}(z) - \tfrac{\lambda}{2}\|u\|^2 + \tfrac{\lambda}{2}\|x - z + u\|^2$$
is $(\beta + \lambda)$-smooth and $(\lambda - \mu)$-strongly convex in $x$.

*Proof.* From $\beta$-smoothness of $\mathcal{L}$ and quadratic penalty $\tfrac{\lambda}{2}\|x - z + u\|^2$, we obtain $(\beta + \lambda)$-smoothness. Since $\lambda > \mu$, the strong convexity term dominates the $\mu$-weak convexity, yielding $(\lambda - \mu)$-strong convexity.

## A.3    CONVERGENCE OF $x$-MINIMIZATION

The $x$-update is given by

$$x^{(t+1)} = x^{(t)} - \eta^{(t)} \Big( \nabla_x \mathcal{L}(G_s^{(t)}, \theta) + \lambda(x^{(t)} - z^{(t)} + u^{(t)}) \Big).$$

Define

$$g^{(t)} := \nabla_x \mathcal{L}(G_s^{(t)}, \theta) + \lambda(x^{(t)} - z^{(t)} + u^{(t)}).$$

By $\beta$-smoothness of $\hat{\mathcal{L}}$, we have

$$\hat{\mathcal{L}}(x^{(t+1)}) \le \hat{\mathcal{L}}(x^{(t)}) - \eta^{(t)} \langle \nabla \hat{\mathcal{L}}(x^{(t)}), g^{(t)} \rangle + \tfrac{\beta(\eta^{(t)})^2}{2} \|g^{(t)}\|^2. \tag{18}$$

**Lemma A.2.** (Projection Consistency) The projection step $z^{(t)} = \Pi_{\mathcal{C}}(x^{(t)} + u^{(t)})$ ensures feasibility of the sparsity constraint $\|z\|_0 \le r|E|$ and preserves boundedness of $\{z^{(t)}\}$.

*Proof.* By non-expansiveness of Euclidean projection,
$$\|z^{(t+1)} - z^{(t)}\| \le \|(x^{(t+1)} - x^{(t)}) + (u^{(t+1)} - u^{(t)})\|.$$

## A.4 Convergence of $\theta$ Updates

The $\theta$-update uses SAM perturbations:

$$\hat{\epsilon}^{(t)} = \rho^{(t)} \frac{\nabla_\theta \mathcal{L}(G_s^{(t)}, \theta^{(t)})}{\|\nabla_\theta \mathcal{L}(G_s^{(t)}, \theta^{(t)})\|}, \qquad \theta^{(t+1)} = \theta^{(t)} - \eta^{(t)} \nabla_\theta \mathcal{L}(G_s^{(t)}, \theta^{(t)} + \hat{\epsilon}^{(t)}).$$

**Lemma A.3.** (Lemma B.1 of (Khanh et al., 2024)) Let $\{a^{(t)}\}, \{b^{(t)}\}, \{c^{(t)}\}$ be nonnegative sequences satisfying

$$a^{(t+1)} - a^{(t)} \leq b^{(t)} a^{(t)} + c^{(t)},$$

with conditions

$$\sum_{t=1}^\infty b^{(t)} = \infty, \qquad \sum_{t=1}^\infty c^{(t)} < \infty, \qquad \sum_{t=1}^\infty b^{(t)} a^{(t)} < \infty.$$

Then $a^{(t)} \to 0$ as $t \to \infty$.

**Lemma A.4.** (Perturbation Stability) Under Assumptions A.2–A.4, the perturbed gradient satisfies

$$\|\nabla_\theta \mathcal{L}(G_s^{(t)}, \theta^{(t)} + \hat{\epsilon}^{(t)}) - \nabla_\theta \mathcal{L}(G_s^{(t)}, \theta^{(t)})\| \leq \beta \rho^{(t)}.$$

*Proof.* By $\beta$-smoothness of $\mathcal{L}$, the deviation due to $\hat{\epsilon}^{(t)}$ is upper bounded by $\beta \|\hat{\epsilon}^{(t)}\| = \beta \rho^{(t)}$.

## A.5 Convergence to Stationary Points

**Theorem A.1.** (Stationarity of Limit Points) Under Assumptions A.1–A.5, the iterates of Algorithm 1 satisfy

$$\nabla_x \hat{\mathcal{L}}(x^{(t)}, z^{(t)}, u^{(t)}, \theta^{(t)}) \to 0, \qquad \nabla_\theta \mathcal{L}(G_s^{(t)}, \theta^{(t)}) \to 0, \quad \text{as } t \to \infty.$$

Thus, every limit point $(\bar{x}, \bar{z}, \bar{u}, \bar{\theta})$ is a stationary point of the augmented Lagrangian with SAM-regularized parameter updates.

*Proof.* From equation 18, we see that $\hat{\mathcal{L}}(x^{(t)})$ decreases up to error terms proportional to $(\eta^{(t)})^2$. By Assumptions A.3–A.4, $\sum_t \eta^{(t)} \rho^{(t)} < \infty$, ensuring bounded cumulative perturbation. For $\theta$, Lemma A.4 guarantees perturbation errors vanish as $\rho^{(t)} \to 0$. Applying Lemma A.3 (Robbins–Siegmund type argument), we obtain

$$\lim_{t \to \infty} \|\nabla_x \hat{\mathcal{L}}(x^{(t)})\| = 0, \qquad \lim_{t \to \infty} \|\nabla_\theta \mathcal{L}(G_s^{(t)}, \theta^{(t)})\| = 0.$$

Therefore, every accumulation point is stationary in both $(x, z, u)$ and $\theta$.

## A.6 Corollaries

**Corollary A.1.** (Expected Convergence) If the gradient is estimated via unbiased stochastic samples with bounded variance, then the expected squared gradient norm satisfies

$$\mathbb{E}\left[\|\nabla \hat{\mathcal{L}}(x^{(t)})\|^2\right] \to 0 \quad \text{as } t \to \infty.$$

*Proof.* This follows directly from Theorem A.1 and the assumption that stochastic gradients are unbiased with bounded variance (Hypothesis A.2). Applying Lemma A.3, we obtain the convergence of expected gradient norms.

**Corollary A.2.** (Convergence Rate) If the step size is chosen as $\eta^{(t)} = \frac{1}{\sqrt{t}}$ and the perturbation radius satisfies $\rho^{(t)} = O(\frac{1}{\sqrt{t}})$, then

$$\min_{1 \leq t \leq T} \mathbb{E}\left[\|\nabla \hat{\mathcal{L}}(x^{(t)})\|^2\right] = O\left(\frac{1}{\sqrt{T}}\right).$$

*Proof.* The rate follows by combining the descent inequality equation 18, bounded perturbation from Lemma A.4, and the standard analysis of diminishing step sizes.

**Scaling with Graph Size.**   An important implication of our analysis is how sparsification interacts with graph size. Suppose the graph has $N$ nodes with average degree $\bar{d}$, so that $|E| \approx N\bar{d}$. For a fixed sparsification ratio $\rho$, the number of preserved edges is $r|E|$. As $N \to \infty$, the redundancy of edges increases, and the variance introduced by random edge removal vanishes:

$$\frac{\text{Var[edge sampling]}}{|E|} \to 0.$$

This provides an intuitive explanation of why our sharpness-aware sparsification benefits become more pronounced on large-scale graphs such as Reddit.

## B   EXPERIMENTAL SETTINGS

| Dataset | #Nodes | #Edges | #Features | #Classes | Split ratio |
|---------|--------|--------|-----------|----------|-------------|
| Cora | 2,708 | 5,429 | 1,433 | 7 | 120/500/1000 |
| Citeseer | 3,327 | 4,732 | 3,703 | 6 | 140/500/1000 |
| PubMed | 19,717 | 44,338 | 500 | 3 | 60/500/1000 |
| ogbn-arxiv | 169,343 | 1,166,243 | 128 | 40 | 54%/18%/28% |
| Reddit | 232,965 | 114,615,892 | 602 | 41 | 66%/10%/24% |

Table 2: Statistics of benchmark datasets.

Table 2 summarizes the datasets used in our experiments, including the number of nodes, edges, features, classes and split ratios. We adopt the public splits from (Yang et al., 2016).

### B.1   IMPLEMENTATION DETAILS

| Hyper-parameter | Value / Search Space |
|-----------------|----------------------|
| Epochs | 200 |
| Learning rate ($\eta$) | 0.001 |
| Learning rate schedule | cosine |
| Weight decay | 0.005 |
| Dropout | 0.5 |
| Hidden units | 128 |
| Attention heads | 8 |
| $\beta$ | 0.2 |
| $\tau$ (Gumbel temperature) | cosine schedule |
| Perturbation radius ($\rho$) | $\{0.01, 0.05, 0.1, 0.2, 0.5, 0.8, 1.0\}$ |
| Dual-update interval ($K$) | $\{1, 2, 5, 10, 20, 40\}$ |
| Penalty parameter ($\lambda$) | $\{0.0001, 0.001, 0.01, 0.1\}$ |

Table 3: Hyperparameter details used for GRAPHSPA

All experiments are implemented in PyTorch (Paszke et al., 2019) and PyTorch Geometric (Fey & Lenssen, 2019), and conducted on four NVIDIA RTX 4090 GPUs (24GB each). Each experiment is repeated with five random seeds, and we report the average accuracy and the standard deviation. We adopt GCN, GAT, GIN, and GraphSAGE as backbone in our experiments. For GCN, we use a two-layer architecture with 128 hidden units, weight decay of 0.005, and dropout rate of 0.5. GAT has two layers with 128 hidden units, and employs 8 attention heads and a dropout rate of 0.5. For GIN, we use a two-layer network with 128 hidden units and dropout rate of 0.5. GraphSAGE is implemented following the configuration with two layers, 128 hidden units, and a dropout rate of 0.5. We adopt a cosine learning rate schedule across all models. In our method, hyperparameters play a role, including the perturbation radius ($\rho$), dual-update interval ($K$), and penalty parameter ($\lambda$). These hyperparameters are tuned via grid search for each dataset, and the final results are reported using the best configuration selected from the search space summarized in Table 3.

# C    COMPUTATION COMPLEXITY ANALYSIS

## C.1    SPARSIFICATION COMPLEXITY

The computational complexity of our self-supervised sparsification framework can be decomposed into three main components: (i) sparsification, (ii) contrastive loss computation, and (iii) encoder training.

**Sparsification stage.**    Each epoch involves parameterizing edge scores $x \in \mathbb{R}^{|E|}$, applying the Gumbel–Sigmoid relaxation, and constructing the normalized sparse adjacency, which requires

$$O(|E|).$$

In addition, every $K$ iterations a projection step is performed at a cost of

$$O(|E| \log |E|),$$

which amortizes to $\frac{1}{K} O(|E| \log |E|)$ per epoch.

**Contrastive loss Computation.**    Constructing the similarity matrix between embeddings $z_1, z_2 \in \mathbb{R}^{N \times d}$ has a complexity of

$$O(N^2 d).$$

When negative sampling or mini-batch contrastive learning is adopted, this reduces to

$$O(Nd).$$

**Encoder training.**    For each forward/backward pass, the GNN encoder requires

$$O(|E|d).$$

Since SAM optimization performs two such passes per epoch, the encoder cost is effectively doubled, though it remains $O(|E|d)$ in asymptotic order.

**Total complexity.**    Putting everything together, the per-epoch complexity is

$$O(|E|d + N^2 d) + \frac{1}{K} O(|E| \log |E|),$$

and for $T$ epochs, the total complexity becomes

$$O\left(T \cdot (|E|d + N^2 d) + \frac{T}{K} |E| \log |E|\right).$$

**Simplification.**    The number of edges can be approximated by the average degree $\bar{d}_{avg}$ as $|E| \approx \frac{1}{2} N \bar{d}_{avg}$. Thus, the edge-related term simplifies to $|E|d \approx N \bar{d}_{avg} d$. For sparse graphs where $\bar{d}_{avg} = O(1)$, we obtain $|E|d = O(Nd)$, showing that the edge cost grows linearly with $N$ and $d$.

**Final complexity.**    After simplification, the dominant cost depends on the loss calculation scheme:

- **Full contrastive learning:** all node pairs are compared, so the $N^2 d$ term dominates, leading to

$$\boxed{O(T \cdot (N^2 d + Nd))}.$$

- **Negative sampling:** only sampled edges are considered, so message passing dominates, giving

$$\boxed{O(T \cdot |E|d)}$$    which simplifies to $O(TNd)$ for sparse graphs.

| Method | Time Complexity | Explanation |
|---|---|---|
| DropEdge | $O(\|E\|)$ | Random edge sampling per epoch: $O(\|E\|)$ |
| Degree | $O(\|E\| \log \|E\|)$ | Degree computation: $O(\|E\|)$ 
 Edge sorting: $O(\|E\| \log \|E\|)$ |
| PageRank | $O(\|E\| \log \|E\|)$ | Power iteration PageRank: $O(K\|E\|)$ 
 Edge scoring: $O(\|E\|)$, sorting: $O(\|E\| \log \|E\|)$ |
| Spectral (ER) | $O(N^3 + \|E\|)$ | Dense Laplacian eigen-decomposition: $O(N^3)$ 
 Effective resistance per edge: $O(\|E\|)$ |
| Shortest-Path | $O(\|E\|(N + \|E\|))$ | For each edge, BFS-based shortest path: $O(N + \|E\|)$ 
 Repeated over all $\|E\|$ edges: $O(\|E\|(N + \|E\|))$ |
| Topo-Cycle | $O(\|E\|^2)$ | Cycle basis extraction: $O(N + \|E\| + \sum_i \|C_i\|)$ 
 Cycle-based edge scoring: $O(\sum_i \|C_i\|) = O(\|E\|C) \leq O(\|E\|^2)$ |
| Topo-Cluster | $O(\|E\|N)$ | Triangle counting per edge: $O(\min(\deg(i), \deg(v)))$ 
 Total: $O(\sum_{(i,j) \in E} \min(\deg(i), \deg(j)))$ 
 $\leq O(\|E\|\Delta) \leq O(\|E\|N)$ |
| GRAPHSPA | $O(TN^2d)$ or $O(T\|E\|d)$ | Subgraph sampling: $O(\|E\|)$ 
 Contrastive loss computation: $O(N^2d)$ or $O(\|E\|d)$ 
 Encoder training: $O(Nd)$ |

Table 4: Time complexity analysis for baseline sparsification methods and GRAPHSPA.

## C.2 SPARSIFICATION COMPLEXITY COMPARISION

Table 4 summarizes the computational complexity of existing sparsification methods compared to GRAPHSPA. Spectral approaches rely on expensive operations such as Laplacian eigendecomposition, shortest-path spanners require repeated BFS expansions for many edges, and topology-based methods depend on cycle or triangle extraction. These procedures incur substantial computational overhead and tend to scale poorly as the number of edges or the structural complexity of the graph increases, making them impractical for large-scale datasets.

In contrast, GRAPHSPA trains the encoder jointly with sparsification, but the dominant cost still comes from the standard GNN forward–backward propagation that all methods share. The additional computations introduced by the augmented Lagrangian module are minimal in practice: the $z$-update involves an $O(\|E\| \log \|E\|)$ sorting step, and the $u$-update requires only simple elementwise operations of complexity $O(\|E\|)$. Moreover, these updates are performed only once every $K$ iterations (we use $K = 20$), so their amortized overhead accounts for less than 3% of the total training time. Although flatness-aware optimization theoretically increases gradient computation, its practical overhead in GRAPHSPA remains limited because perturbations are applied only to the encoder parameters. Empirically, the wall-clock time increases by approximately 1.4–1.6× rather than the full 2× expected from theory. Furthermore, optimizing toward flatter minima improves robustness to residual noise and reduces the number of training epochs required by roughly 20–30%, compensating for part of the additional cost.

# D ABLATION STUDIES

## D.1 IMPACT OF FLATNESS-AWARE TRAINING DURING SPARSIFICATION

| Method | $r = 0.9$ | 0.8 | 0.7 | 0.6 | 0.5 |
|---|---|---|---|---|---|
| GRAPHSPA (Frozen) | $76.60_{\pm 0.21}$ | $76.06_{\pm 2.80}$ | $76.46_{\pm 1.11}$ | $75.16_{\pm 1.80}$ | $74.44_{\pm 1.24}$ |
| GRAPHSPA (w/ Adam) | $76.52_{\pm 1.49}$ | $75.90_{\pm 2.52}$ | $76.66_{\pm 1.28}$ | $75.58_{\pm 1.46}$ | $74.86_{\pm 1.02}$ |
| **GRAPHSPA (w/ SAM)** | $\mathbf{78.34}_{\pm 0.76}$ | $\mathbf{77.52}_{\pm 1.04}$ | $\mathbf{77.10}_{\pm 0.74}$ | $\mathbf{76.72}_{\pm 0.45}$ | $\mathbf{77.26}_{\pm 1.13}$ |

Table 5: Ablation study on encoder training during sparsification on Pubmed using GCN. Results are reported as mean ± std over five random seeds.

Table 5 presents the ablation results on the Pubmed dataset, comparing three settings: Frozen Encoder, Adam, and SAM. The results highlight that encoder training during sparsification is crucial for achieving good generalization and robustness to noisy edges. Freezing the encoder significantly de-

grades accuracy since the embeddings cannot adapt to the evolving sparse graph structure. Training with Adam provides moderate results but is less robust across edge ratios. In contrast, SAM consistently achieves the best performance, demonstrating that flatness-aware optimization enhances stability during training and yields more reliable performance under varying sparsity levels.

## D.2 PERFORMANCE ON HETEROPHILIC GRAPHS

To further evaluate the generality of GRAPHSPA, we conduct experiments on two widely used heterophilic benchmark datasets: Actor (Pei et al., 2020) and Chameleon (Rozemberczki & Sarkar, 2021). Unlike homophilous citation networks, these graphs exhibit low homophily ratios, meaning that edges frequently connect nodes from different classes. This makes learning substantially more challenging for most GNNs, as local neighborhoods do not reliably encode label information.

| Dataset | #Nodes | #Edges | #Features | #Classes | Homophily Rate |
|---------|--------|--------|-----------|----------|----------------|
| Actor | 7,600 | 26,752 | 932 | 5 | 0.22 |
| Chameleon | 2,277 | 36,101 | 2,325 | 5 | 0.23 |

Table 6: Statistics of heterophilic datasets.

Table 6 provides comprehensive statistics of the datasets used in our experiments, including the number of nodes, edges, classes, features and homophily rate.

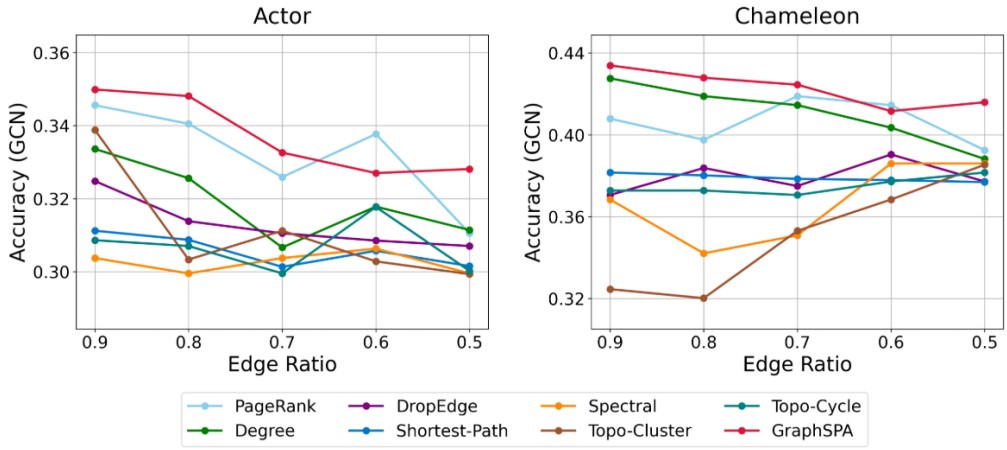

Figure 4: Node classification accuracy on heterophilic datasets Actor/Chameleon using GCN under various sparsity ratios. Results are reported as mean over five random seeds, and statistical significance is validated with paired t-tests $p < 0.05$.

As shown in Figure 4, GRAPHSPA achieves the highest accuracy across almost all sparsity levels on both heterophilic datasets. In contrast, traditional sparsification methods primarily preserve structural properties such as degree, local topology, or shortest path distances, but these properties often do not align with the semantic relationships in heterophilic graphs. As a result, structure-preserving sparsifiers tend to retain edges that do not contribute to meaningful representation learning. GRAPHSPA instead retains edges that preserve representation-level signals, including node features, multi-hop dependencies, and relationships in the embedding space. This characteristic enables GRAPHSPA to remain effective even when nodes from different classes are frequently connected. Overall, the results demonstrate that GRAPHSPA is highly robust in heterophilic scenarios and can successfully sparsify graphs where conventional notions of structural similarity are unreliable. This highlights its practical applicability to real-world networks where connections between semantically dissimilar entities naturally arise.

### D.3 INFLUENCE OF THE INFONCE NEGATIVE SAMPLING TEMPERATURE $\beta$

The InfoNCE temperature $\beta$ controls the sharpness of the negative sampling distribution and there-fore affects the magnitude of the mutual information gradient. In GRAPHSPA, $\beta$ appears only inside the MI loss term within the augmented Lagrangian formulation in Equation (12). As a result, $\beta$ scales the magnitude of the MI gradient but does not alter the structure of the objective, the sparsity constraint, or the direction of the optimization process. A key property of GRAPHSPA is that the edge scores are not determined by a single MI gradient update. The sparsity projection variable $z$ enforces the target sparsity level, while the dual variable $u$ accumulates the deviation between $x$ and $z$ and corrects it over subsequent iterations. These augmented Lagrangian dynamics operate inde-pendently of $\beta$, meaning that most of the variation induced by scaling the MI gradient is absorbed by the penalty term, keeping the trajectory of the edge scores stable. To empirically verify this effect, we conduct sensitivity experiments on Cora, Citeseer, and Pubmed using a GCN backbone and a 70% edge retention ratio. We vary the temperature parameter over $\beta \in \{0.2, 0.5, 0.8, 1.0, 1.5\}$ and report the average accuracy over five random seeds. The results are summarized in Table 7.

| Dataset | $\beta = 0.2$ | 0.5 | 0.8 | 1.0 | 1.5 |
|---|---|---|---|---|---|
| Cora | $78.48 \pm 0.88$ | $78.98 \pm 0.94$ | $78.88 \pm 1.42$ | $78.56 \pm 0.89$ | $78.46 \pm 1.14$ |
| Citeseer | $67.46 \pm 1.40$ | $67.72 \pm 2.53$ | $67.78 \pm 1.39$ | $67.44 \pm 1.69$ | $67.36 \pm 4.05$ |
| Pubmed | $76.74 \pm 1.10$ | $76.72 \pm 0.90$ | $76.68 \pm 1.26$ | $76.64 \pm 1.21$ | $77.06 \pm 0.72$ |

Table 7: Sensitivity of GRAPHSPA to the InfoNCE temperature $\beta$.

Across all datasets, the differences in accuracy remain within approximately 0.4 percent, which is comparable to natural seed variance. These results confirm that $\beta$ primarily adjusts the scale of the MI gradient, while the augmented Lagrangian penalty terms regulate the edge-score updates and maintain stability throughout training. Overall, GRAPHSPA shows strong robustness to the choice of the temperature $\beta$, and variations in this parameter have minimal impact on the final sparsified graph and downstream performance.

### D.4 SENSITIVITY TO HYPERPARAMETER $\rho$

The perturbation radius $\rho$ introduced by SAM is a critical hyperparameter that controls the extent of parameter perturbations during optimization. Choosing an appropriate $\rho$ is essential for balancing robustness and training stability.

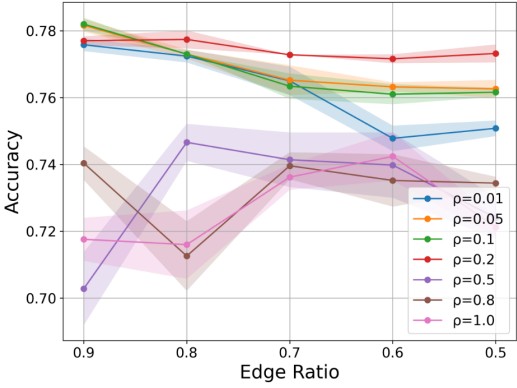

Figure 5: Sensitivity of performance to the SAM perturbation radius $\rho$ on Pubmed. Results are reported as mean ± std over five random seeds.

- **Large $\rho$:** When $\rho$ is large, the optimizer explores flatter regions in the loss landscape, which can potentially improve generalization and robustness. However, overly large perturbations may destabilize training or hinder convergence, leading to degraded performance.
- **Small $\rho$:** When $\rho$ is too small, it may result in limited robustness gains, as the perturbations are not sufficient to promote significant flatness in the parameter space.

We conducted experiments by varying $\rho \in \{0.01, 0.05, 0.1, 0.2, 0.5, 0.8, 1.0\}$ to evaluate its impact on performance. Figure 5 illustrates the test accuracy across different edge ratios. Small $\rho$ (e.g., 0.01) provides only minor improvements, while very large $\rho$ (e.g., 0.5 or above) causes unstable training and significant degradation. An intermediate range (e.g., $\rho = 0.1$ or $\rho = 0.2$) yields the best trade-off between robustness and stability.

## D.5 EFFECTS OF $\tau$ SCHEDULING

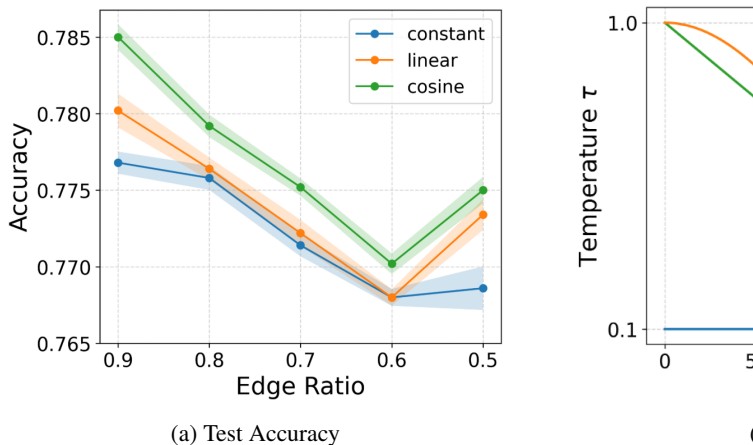

(a) Test Accuracy        (b) Schedules

Figure 6: Effects of $\tau$ scheduling. (a) Sparsification accuracy across different sparsification ratios for constant $\tau$ compared with linear and cosine schedules. (b) Illustration of $\tau$ scheduling strategies, where the cosine schedule maintains a higher $\tau$ in the early phase and decreases later for exploitation.

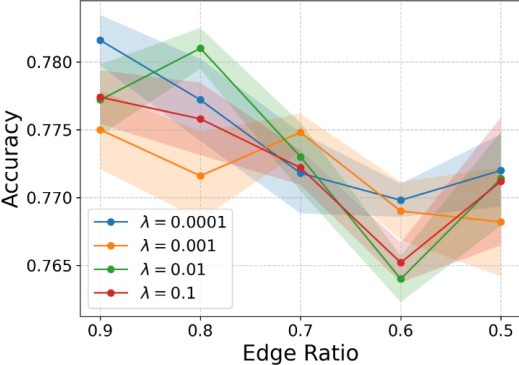

Figure 7: Effect of different penalty parameters $\lambda$ on validation accuracy on Pubmed under varying edge ratios. Results are reported as mean ± std over five random seeds.

In addition to fixed $\tau$, we investigate different scheduling strategies to dynamically adjust the temperature during training. As shown in Figure 6, we compare sparsification performance under different $\tau$ settings. (a) demonstrates the effect of constant $\tau$ versus linear and cosine scheduling on sparsification accuracy across various sparsification ratios. (b) illustrates the scheduling dynamics of $\tau$, where the cosine schedule starts with a relatively higher $\tau$ to encourage exploration of diverse subgraphs through broader edge distributions, and then gradually decays to enhance exploitation in the later phase. This gradual transition from exploration to exploitation explains why cosine scheduling consistently achieves better performance compared to both constant and linear schedules.

## D.6 EFFECTS OF PENALTY PARAMETER $\lambda$

We investigate the effect of different choices of the penalty parameter $\lambda$ on accuracy across various edge ratios. Figure 7 reports the average accuracy with standard deviation for $\lambda \in$

$\{10^{-4}, 10^{-3}, 10^{-2}, 10^{-1}\}$ under edge ratios ranging from $0.9$ to $0.5$. We observe that larger values of $\lambda$ such as $10^{-1}$ generally alleviate the accuracy drop at moderate edge ratios but degrade the performance when the sparsification becomes more aggressive. In contrast, smaller values such as $\lambda = 10^{-4}$ yield competitive performance at higher edge ratios but fail to stabilize under heavier sparsification. This highlights the trade-off between enforcing the sparsity constraint more strongly via larger $\lambda$ and preserving model accuracy under different sparsity levels. In particular, while $\lambda = 10^{-2}$ achieves the highest performance around edge ratio $0.8$, its accuracy decreases significantly at $0.6$, indicating that the choice of $\lambda$ must be carefully balanced depending on the target sparsity.

