# OpenReview forum: "GraphSpa: Self-supervised Graph Sparsification for robust generalization"
_ICLR.cc/2026/Conference — ICLR 2026 Conference Withdrawn Submission_

### Official Review · Reviewer_8WsS · 2025-10-25

**Soundness:** 2
**Presentation:** 2
**Contribution:** 2
**Rating:** 2
**Confidence:** 4

**Summary:**

The paper studies self-supervised sparsification of graphs for node classification. They first use Gumbel-softmax method to sample important edges in a differentiable way and then incorporate the robust training method of sharpness-aware minimization into the training process. Optimization process via the augmented Lagrangian relaxation. Experiments are conducted on classic GNNs with performance comparison with unsupervised baselines such as DropEdge.

**Strengths:**

Strengths:
1. The paper studies the interesting problem of constructing graph sparsifiers in a self-supervised way. It can be applied to label-free settings and might have wider applicability than supervised algorithms.
2. The paper is easy to follow and explains the key ideas and technical details clearly.

**Weaknesses:**

Weaknesses:
1. While there seems limited study on self-supervised graph sparsification, traditional graph sparsification in graph theory can be considered as unsupervised. They aim to preserve some important graph properties in the original graph, such as spectral properties, random walk properties, and shortest-path distances. The related work largely ignore these existing works, and some of them, e.g., [a,c], could be comparable to the current method. Some supervised sparsification methods, e.g., [b], are also missing. The authors may want to perform a more comprehensive review on existing work and better position the current paper.

References:

[a] Abd Errahmane Kiouche, Julien Baste, Mohammed Haddad, Hamida Seba, and Angela Bonifati. 2024. Neighborhood-Preserving Graph Sparsification. Proc. VLDB Endow. 17, 13 (September 2024), 4853–4866. https://doi.org/10.14778/3704965.3704988

[b] Cheng Zheng, Bo Zong, Wei Cheng, Dongjin Song, Jingchao Ni, Wenchao Yu, Haifeng Chen, and Wei Wang. 2020. Robust graph representation learning via neural sparsification. In Proceedings of the 37th International Conference on Machine Learning (ICML'20), Vol. 119. JMLR.org, Article 1062, 11458–11468.

[c] Sadhanala, V., Wang, Y.-X., and Tibshirani, R. J. Graph sparsification approaches for Laplacian smoothing. In Proceedings of AISTATS Conference, pp. 1250–1259, 2016.

2. The techniques used in this paper are quite typical and I am not feeling excited about the proposed method. The Gumbel-Softmax has been widely used in edge sampling, e.g., in [b] and the flatness-aware training [21] is also well-established. While the authors propose a complete framework, each part of the algorithm is based on a typical method, limiting the overall novelty of the technique.
3. It would be best to involve additional and diverse benchmark datasets in the experimental study.

**Questions:**

Does the method apply to graph classification and link prediction tasks and inductive learning setting? How about directed graphs?

---

> ### Author Response · Authors · 2025-11-21
>
> We sincerely appreciate the reviewer’s thoughtful evaluation and the recognition of the core ideas and motivations underlying our work. We make clarifications to specific comments below.
>
> > #### **W2. Overall novelty of GRAPHSPA**
>
> Thank you for the reviewer’s comments. The reviewer pointed out that each component of our framework relies on techniques that are widely used independently in prior work. We appreciate this observation, and we also agree that each individual technique can appear standard when viewed in isolation. However, as `PDrR S1` and `NpUB S1` both emphasized, our framework is not a simple aggregation of independent techniques. MI-based self-supervision, augmented Lagrangian optimization, and SAM perturbation are not placed side-by-side; rather, they are integrated such that each component compensates for the limitations of the others, collectively ensuring the convergence and stability of the entire algorithm. For example, methods such as Gumbel-softmax play only a secondary role and are not central to the core mechanism of our approach.
>
> Our work addresses two fundamental challenges that cannot be solved by graph sparsification alone:
>
> - First, we require a self-supervised mechanism that performs sparsification using only graph structure, without relying on labels.
>
> - Second, we must address the fact that residual noisy edges that remain after sparsification often make GNNs more vulnerable.
>
> To solve these two challenges simultaneously, one must reconcile two inherently conflicting objectives: sparsity and flatness. The contribution of our work is that we unify these conflicting goals within a single optimization framework whose convergence is theoretically guaranteed. However, attempting to satisfy both objectives at once can cause gradients to oscillate or updates to diverge. Our framework is deliberately designed so that the MI objective, augmented Lagrangian, and SAM perturbation interact without conflict and stabilize one another.
>
> The convergence analysis in Supplementary Materials A is provided to theoretically guarantee the stability of this integrated structure. This analysis is not a mere intuitive argument; under Assumptions A.1–A.5, we rigorously prove that the augmented Lagrangian, projection operator, and SAM perturbation do not interfere with each other.
>
> The augmented Lagrangian enables stable optimization of sparsity-constrained problems in a continuous domain. As shown in Lemma A.1, it is $(\beta + \lambda)$-smooth and $(\lambda - \mu)$-strongly convex, which provides the essential structure needed for convergence under non-linear and non-smooth constraints. The dual variable corrects constraint violation accumulatively, and the projection operator plays a crucial role in keeping $z$ aligned with the target sparsity level. Lemma A.2 demonstrates that this projection operator is non-expansive and ensures the boundedness of the sequence ${ z_t^{(k)} }$, preventing edge scores from exploding under gradient noise or perturbation.
>
> SAM perturbation may potentially destabilize the $\theta$-gradient, but Lemma A.4 (Perturbation Stability) shows that the gradient deviation term $\beta \rho(t)$ induced by SAM decreases over time. With diminishing $\rho(t)$, the perturbation error converges to zero, guaranteeing the stability of the $\theta$-update.
>
> To reiterate, the key novelty of our work does not lie in the use of any individual technique. Rather, our main contribution lies in simultaneously preserving structural information in a label-free setting while mitigating residual noise, by unifying sparsity and flatness objectives within a single optimization framework whose convergence is theoretically ensured. We believe this integrated stability clearly distinguishes our approach from prior sparsification or flatness-based methods.

---

> ### Author Response · Authors · 2025-11-21
>
> > #### **W1. Lack of comparison to sparsification baselines**
>
> Thank you for highlighting the need for broader sparsification comparisons and for recommending related research. In preparing the revised version, we carefully reviewed the methods suggested by the reviewer, including [a], [b], and [c], and expanded our comparison to include a wider set of sparsifiers that reflect the major sparsification objectives studied in the literature.
>
> Although some of the referenced methods (e.g., [a], [c]) do not provide official implementations, making exact reproduction difficult, we ensured that the sparsification objectives highlighted in prior work are faithfully represented. To this end, we incorporated five additional sparsifiers, each designed to preserve a distinct structural aspect of the original graph. The included sparsifiers aim to preserve the following properties:
>
>  - **Spectral preserving sparsifier**: approximates the graph Laplacian quadratic form to maintain the global smoothness and energy structure.
>
>  - **Shortest-path preserving sparsifier**: preserves pairwise distances and shortest-path lengths through path-based pruning.
>
>  - **Random-walk preserving sparsifier**: maintains random-walk transition probabilities and commute-time characteristics.
>
>  - **Topology-cycle preserving sparsifier**: preserves cycle basis, motifs, and topological loop structures while removing redundant edges.
>  - **Topology-clustering based sparsifier**: retains essential intra-community relationships while pruning low-importance connections.
>
>
> | Ratio $r$ | Method           | Cora            | Citeseer         | Pubmed           |
> |---:|--------|----|---|----|
> | 0.9 | Random Walk      | 0.8020 ± 0.0077       | 0.6912 ± 0.0114        | 0.7730 ± 0.0033        |
> |      | Shortest Path     | 0.7196 ± 0.0054       | 0.6256 ± 0.0065        | 0.7502 ± 0.0024        |
> |      | Spectral          | 0.7998 ± 0.0076       | 0.6852 ± 0.0066        | 0.7744 ± 0.0014        |
> |      | Topo-Clustering   | 0.7972 ± 0.0101       | 0.6786 ± 0.0149        | 0.7652 ± 0.0059        |
> |      | Topo-Cycle        | 0.7894 ± 0.0087       | 0.6898 ± 0.0063        | 0.7770 ± 0.0109        |
> |      | GRAPHSPA          | **0.8128 ± 0.0038**       | **0.6916 ± 0.0111**        | **0.7834 ± 0.0083**        |
> | 0.8 | Random Walk      | 0.7880 ± 0.0079       | 0.6776 ± 0.0100        | 0.7674 ± 0.0014        |
> |      | Shortest Path     | 0.7182 ± 0.0143       | 0.6278 ± 0.0154        | 0.7522 ± 0.0039        |
> |      | Spectral          | 0.7950 ± 0.0101       | 0.6870 ± 0.0049        | 0.7740 ± 0.0030        |
> |      | Topo-Clustering   | 0.7852 ± 0.0078       | 0.6710 ± 0.0141        | 0.7578 ± 0.0062        |
> |      | Topo-Cycle        | 0.7830 ± 0.0052       | 0.6850 ± 0.0048        | 0.7616 ± 0.0047        |
> |      | GRAPHSPA          | **0.8056 ± 0.0096**       | **0.6880 ± 0.0115**        | **0.7752 ± 0.0108**        |
> | 0.7 | Random Walk      | 0.7688 ± 0.0049       | 0.6790 ± 0.0091        | 0.7674 ± 0.0073        |
> |      | Shortest Path     | 0.7166 ± 0.0156       | 0.6182 ± 0.0102        | 0.7448 ± 0.0046        |
> |      | Spectral          | 0.7934 ± 0.0020       | 0.6812 ± 0.0046        | 0.7694 ± 0.0022        |
> |      | Topo-Clustering   | 0.7378 ± 0.0137       | 0.6694 ± 0.0054        | 0.7564 ± 0.0044        |
> |      | Topo-Cycle        | 0.7656 ± 0.0060       | 0.6684 ± 0.0035        | 0.7640 ± 0.0099        |
> |      | GRAPHSPA          | **0.7906 ± 0.0142**       | **0.6816 ± 0.0102**        | **0.7710 ± 0.0086**        |
> | 0.6 | Random Walk      | 0.7688 ± 0.0054       | 0.6654 ± 0.0195        | 0.7634 ± 0.0035        |
> |      | Shortest Path     | 0.7272 ± 0.0078       | 0.6260 ± 0.0184        | 0.7528 ± 0.0036        |
> |      | Spectral          | 0.7710 ± 0.0057       | 0.6798 ± 0.0128        | 0.7692 ± 0.0042        |
> |      | Topo-Clustering   | 0.7252 ± 0.0190       | 0.6570 ± 0.0028        | 0.7526 ± 0.0101        |
> |      | Topo-Cycle        | 0.7592 ± 0.0092       | 0.6520 ± 0.0080        | 0.7482 ± 0.0113        |
> |      | GRAPHSPA          | **0.7872 ± 0.0086**       | **0.6734 ± 0.0145**        | **0.7722 ± 0.0050**        |
> | 0.5 | Random Walk      | 0.7506 ± 0.0050       | 0.6612 ± 0.0114        | 0.7424 ± 0.0178        |
> |      | Shortest Path     | 0.7256 ± 0.0092       | 0.6222 ± 0.0149        | 0.7530 ± 0.0080        |
> |      | Spectral          | 0.7564 ± 0.0037       | 0.6548 ± 0.0140        | 0.7720 ± 0.0040        |
> |      | Topo-Clustering   | 0.7096 ± 0.0099       | 0.6378 ± 0.0048        | 0.7356 ± 0.0063        |
> |      | Topo-Cycle        | 0.7484 ± 0.0069       | 0.6580 ± 0.0103        | 0.7440 ± 0.0076        |
> |      | GRAPHSPA          | **0.7738 ± 0.0088**       | **0.6668 ± 0.0097**        | **0.7726 ± 0.0126**        |
>
>
> We implemented all five sparsifiers and included them in our experiments. Across all datasets, GRAPHSPA consistently outperforms these sparsifiers.

---

> ### Author Response · Authors · 2025-11-21
>
> > #### **W3. Need for more diverse benchmark datasets**
>
> Thank you for suggesting that a wider and more diverse set of benchmark datasets should be included. We completely agree with this point, and we substantially expanded the empirical evaluation in the revised version to reflect your feedback.
>
> In addition to the homophilic datasets used in the original submission (Cora, Citeseer, Pubmed), we conducted experiments on two heterophilic datasets, Actor and Chameleon. Unlike homophilic graphs, where connected nodes tend to share similar labels or feature patterns, heterophilic graphs exhibit the opposite structure: nodes connected by edges often have dissimilar or even conflicting labels and features. This structural heterogeneity is known to disrupt many traditional sparsifiers because sparsification may easily distort already weak or inverted feature–edge relationships. Despite this challenging setting, our results show that GRAPHSPA consistently outperforms alternative sparsifiers across almost all sparsity levels. The results on Actor and Chameleon are listed below.
>
> | Ratio $r$ | Method          | Actor               | Chameleon           |
> |---------:|------------------|---------------------|----------------------|
> | 0.9     | PageRank        | 0.3105 ± 0.0056     | 0.3925 ± 0.0119     |
> |         | Degree          | 0.3114 ± 0.0024     | 0.3882 ± 0.0184     |
> |         | DropEdge        | 0.3070 ± 0.0010     | 0.3772 ± 0.0036     |
> |         | Random-Walk     | 0.3147 ± 0.0052     | 0.3662 ± 0.0080     |
> |         | Shortest-Path   | 0.3015 ± 0.0122     | 0.3769 ± 0.0194     |
> |         | Spectral        | 0.2995 ± 0.0010     | 0.3860 ± 0.0174     |
> |         | Topo-Clustering | 0.2993 ± 0.0133     | 0.3854 ± 0.0086     |
> |         | Topo-Cycle      | 0.3002 ± 0.0048     | 0.3816 ± 0.0129     |
> |         | GRAPHSPA    | **0.3281 ± 0.0019** | **0.4159 ± 0.0091** |
> | 0.8     | PageRank        | **0.3377 ± 0.0036**     | **0.4145 ± 0.0105**     |
> |         | Degree          | 0.3178 ± 0.0034     | 0.4035 ± 0.0187     |
> |         | DropEdge        | 0.3085 ± 0.0045     | 0.3904 ± 0.0130     |
> |         | Random-Walk     | 0.2997 ± 0.0032     | 0.3750 ± 0.0106     |
> |         | Shortest-Path   | 0.3057 ± 0.0176     | 0.3779 ± 0.0099     |
> |         | Spectral        | 0.3063 ± 0.0010     | 0.3860 ± 0.0199     |
> |         | Topo-Clustering | 0.3028 ± 0.0061     | 0.3684 ± 0.0081     |
> |         | Topo-Cycle      | 0.3177 ± 0.0024     | 0.3772 ± 0.0110     |
> |         | GRAPHSPA    | 0.3270 ± 0.0083 | 0.4116 ± 0.0051 |
> | 0.7     | PageRank        | 0.3259 ± 0.0034     | 0.4189 ± 0.0112     |
> |         | Degree          | 0.3066 ± 0.0159     | 0.4145 ± 0.0053     |
> |         | DropEdge        | 0.3105 ± 0.0116     | 0.3750 ± 0.0051     |
> |         | Random-Walk     | 0.2947 ± 0.0017     | 0.3531 ± 0.0081     |
> |         | Shortest-Path   | 0.3013 ± 0.0143     | 0.3785 ± 0.0168     |
> |         | Spectral        | 0.3037 ± 0.0024     | 0.3509 ± 0.0120     |
> |         | Topo-Clustering | 0.3112 ± 0.0154     | 0.3531 ± 0.0084     |
> |         | Topo-Cycle      | 0.2995 ± 0.0024     | 0.3706 ± 0.0194     |
> |         | GRAPHSPA    | **0.3326 ± 0.0068** | **0.4245 ± 0.0066** |
> | 0.6     | PageRank        | 0.3405 ± 0.0121     | 0.3976 ± 0.0024     |
> |         | Degree          | 0.3256 ± 0.0014     | 0.4189 ± 0.0081     |
> |         | DropEdge        | 0.3138 ± 0.0171     | 0.3838 ± 0.0035     |
> |         | Random-Walk     | 0.3066 ± 0.0169     | 0.3772 ± 0.0093     |
> |         | Shortest-Path   | 0.3087 ± 0.0138     | 0.3802 ± 0.0097     |
> |         | Spectral        | 0.2995 ± 0.0100     | 0.3421 ± 0.0170     |
> |         | Topo-Clustering | 0.3033 ± 0.0038     | 0.3202 ± 0.0109     |
> |         | Topo-Cycle      | 0.3070 ± 0.0177     | 0.3728 ± 0.0051     |
> |         | GRAPHSPA    | **0.3481 ± 0.0053** | **0.4279 ± 0.0245** |
> | 0.5     | PageRank        | 0.3456 ± 0.0122     | 0.4079 ± 0.0088     |
> |         | Degree          | 0.3336 ± 0.0156     | 0.4276 ± 0.0092     |
> |         | DropEdge        | 0.3248 ± 0.0155     | 0.3706 ± 0.0130     |
> |         | Random-Walk     | 0.3000 ± 0.0112     | 0.4079 ± 0.0069     |
> |         | Shortest-Path   | 0.3112 ± 0.0073     | 0.3816 ± 0.0064     |
> |         | Spectral        | 0.3037 ± 0.0090     | 0.3684 ± 0.0110     |
> |         | Topo-Clustering | 0.3388 ± 0.0161     | 0.3246 ± 0.0114     |
> |         | Topo-Cycle      | 0.3086 ± 0.0184     | 0.3728 ± 0.0114     |
> |         | GRAPHSPA    | **0.3499 ± 0.0075** | **0.4339 ± 0.0231** |
>
> Overall, these results demonstrate that GRAPHSPA remains robust even in heterophilic environments where many traditional sparsifiers degrade significantly. By relying on a representation-driven MI objective rather than hand-crafted structural preservation rules, GRAPHSPA is able to retain edges that meaningfully support encoder representations, allowing it to generalize across graph types with very different structural characteristics.

---

> ### Author Response · Authors · 2025-11-21
>
> > #### **W3. Need for more diverse benchmark datasets**
>
> We also extended the evaluation to two large-scale datasets, Reddit and ogbn-arxiv. These graphs contain millions to hundreds of millions of edges, so they provide an important stress test for sparsification methods. All experiments were conducted under identical conditions using a GraphSAGE encoder and four NVIDIA RTX 4090 GPUs.
>
> Before presenting the large-scale results, we would like to point out an important practical limitation of many classical sparsifiers. Methods that aim to preserve specific structural properties, such as spectral structure, random-walk behavior, shortest-path distances, or topological cycle information, often require heavy computation. Several of these methods failed with time-limit errors on Reddit and ogbn-arxiv. For example, some topology preserving sparsifiers require complexity on the order of $O(|E| d_{\text{avg}}^{4}))$ (where $d_{\text{avg}}$ is the average node degree), which is not feasible for graphs containing millions of edges.
>
> In contrast, GRAPHSPA performs sparsification in time proportional to the number of edges. This makes it possible to run GRAPHSPA efficiently even on very large graphs. For this reason, the large-scale evaluation includes only the sparsifiers that could be executed within a reasonable time budget. The detailed results are presented below.
> | Ratio $r$ | Method      | ogbn-arxiv           | Reddit               |
> |--------:|--|--|----|
> | 0.9     | PageRank    | 0.6948 ± 0.0152      | 0.9360 ± 0.0189      |
> |         | Degree      | 0.7056 ± 0.0113      | 0.9433 ± 0.0072      |
> |         | DropEdge    | 0.6889 ± 0.0068      | 0.9360 ± 0.0105      |
> |         | Random-Walk | 0.6902 ± 0.0099      | 0.9222 ± 0.0156      |
> |         | GraphSPA    | **0.7282 ± 0.0046**      | **0.9618 ± 0.0042**      |
> | 0.8     | PageRank    | 0.6894 ± 0.0075      | 0.9341 ± 0.0061      |
> |         | Degree      | 0.7001 ± 0.0054      | 0.9411 ± 0.0137      |
> |         | DropEdge    | 0.6920 ± 0.0149      | 0.9389 ± 0.0058      |
> |         | Random-Walk | 0.6847 ± 0.0174      | 0.9200 ± 0.0049      |
> |         | GraphSPA    | **0.7215 ± 0.0031**      | **0.9597 ± 0.0036**      |
> | 0.7     | PageRank    | 0.6820 ± 0.0124      | 0.9320 ± 0.0094      |
> |         | Degree      | 0.6944 ± 0.0168      | 0.9394 ± 0.0048      |
> |         | DropEdge    | 0.6852 ± 0.0083      | 0.9347 ± 0.0141      |
> |         | Random-Walk | 0.6785 ± 0.0038      | 0.9183 ± 0.0163      |
> |         | GraphSPA    | **0.7157 ± 0.0044**      | **0.9589 ± 0.0033**      |
> | 0.6     | PageRank    | 0.6731 ± 0.0097      | 0.9298 ± 0.0175      |
> |         | Degree      | 0.6839 ± 0.0072      | 0.9369 ± 0.0112      |
> |         | DropEdge    | 0.6768 ± 0.0047      | 0.9322 ± 0.0039      |
> |         | Random-Walk | 0.6679 ± 0.0191      | 0.9155 ± 0.0110      |
> |         | GraphSPA    | **0.7084 ± 0.0041**      | **0.9558 ± 0.0068**      |
> | 0.5     | PageRank    | 0.6618 ± 0.0180      | 0.9251 ± 0.0087      |
> |         | Degree      | 0.6723 ± 0.0135      | 0.9321 ± 0.0065      |
> |         | DropEdge    | 0.6642 ± 0.0091      | 0.9278 ± 0.0154      |
> |         | Random-Walk | 0.6575 ± 0.0104      | 0.9114 ± 0.0198      |
> |         | GraphSPA    | **0.6964 ± 0.0038**      | **0.9524 ± 0.0045**      |
>
> ---
>
> We additionally provide the statistics of all newly added datasets used in the extended evaluation.
>
>
> | Dataset     | #Nodes  | #Edges        | #Features | #Classes |
> |-------------|---------|---------------:|-----------:|---------:|
> | Actor       | 7,600   | 26,752         | 932        | 5        |
> | Chameleon   | 2,277   | 36,101         | 2,325      | 5        |
> | ogbn-arxiv  | 169,343 | 1,166,243      | 128        | 40       |
> | Reddit      | 232,965 | 114,615,982    | 602        | 41       |
>
> We carefully incorporated the reviewer’s suggestion and significantly broadened the experimental evaluation by including both structurally challenging heterophilic datasets and large-scale real-world graphs. Across all newly added benchmarks, GRAPHSPA consistently demonstrated strong performance, robustness, and scalability.

---

> > ### Author Response · Authors · 2025-11-21
> >
> > > #### **Q1. Does the method apply to graph classification and link prediction tasks and inductive learning setting? How about directed graphs?**
> >
> > We thank the reviewer for raising an important question regarding the applicability of GRAPHSPA to graph classification, link prediction, inductive learning, and directed graphs. GRAPHSPA is formulated as an edge-level sparsification framework for a single graph, and its compatibility with each task depends on how well the task’s assumptions align with this objective.
> >
> > For graph classification, GRAPHSPA can be applied without difficulty. Since graph classification processes multiple independent graphs and predicts a label for each, one may simply sparsify each graph using GRAPHSPA and then perform graph-level pooling and classification. Because this task does not require preserving all edges, sparsification does not conflict with the overall task formulation.
> >
> > In contrast, link prediction uses the presence or absence of an edge as the supervision signal itself. Because GRAPHSPA intentionally removes task-irrelevant edges during sparsification, applying it in this setting would directly modify the ground-truth labels, making the task definition and the sparsification process fundamentally incompatible. Therefore, GRAPHSPA is not directly suitable for link prediction in its current form.
> >
> > For inductive learning, the framework is fully compatible. GRAPHSPA operates through the encoder, and when combined with inductive GNNs such as GraphSAGE, it can generalize to unseen nodes or subgraphs. The Reddit dataset used in our experiments is a standard inductive benchmark, demonstrating that GRAPHSPA functions reliably under inductive settings.
> >
> > Directed graphs can also be supported with minimal extension. Directionality can be preserved by treating $x_{ij}$ and $x_{ji}\$ as separate sparsification variables and learning them independently. Only the MI objective requires a minor adjustment to follow a directed message-passing formulation. All other components, including augmented Lagrangian updates, and SAM perturbation, remain unchanged. Thus, GRAPHSPA can be extended to directed graphs without altering the core architecture.
> >
> > We sincerely appreciate the reviewer’s thoughtful question and the opportunity to clarify the scope of GRAPHSPA. We will incorporate these discussions, along with experimental results and additional analysis mentioned above, into the revised version of the manuscript.

---

> > > ### Comment · Reviewer_8WsS · 2025-11-23
> > >
> > > Thank you for the detailed response. While the added experiments could have been in the original submission and made it stronger, the authors need to carefully provide justifications on the selection of baseline methods, e.g., are they top-performers in prior studies such as "Demystifying Graph Sparsification Algorithms in Graph Properties Preservation". The tuning of the learning methods should also be made very clear: I'm curious why the learning-based method DropEdge almost always perform worse than the Degree, which is counterintuitive.
> > >
> > > While I am increasing my score to 4, I'm still concerned that the overall novelty of the technique is limited, as described in my review. Extensive experiments would definitely help strength the work but the experiments in the original submission are quite weak.  The substantial modifications to the original submission would require another serious pass of reviews before it's fully ready for publication.

---

> > > > ### Author Response · Authors · 2025-11-26
> > > >
> > > > Thank you for the reviewer’s thoughtful comments. We would like to provide a clearer explanation addressing the points you mentioned.
> > > >
> > > > > #### **Justifications on the selection of baseline methods**
> > > >
> > > > The paper “Demystifying Graph Sparsification Algorithms in Graph Properties Preservation” serves as an important reference that systematically compares a wide range of classical sparsifiers. This work includes Random, Degree, Spanning Forest, K-Neighbor, ER, and several other traditional sparsification methods. However, as explicitly noted in that paper (Section 3.2), several of these sparsifiers cannot directly control the sparsity ratio (e.g., Spanning Forest, t-Spanner), while others support only coarse-grained control over sparsity levels (e.g., K-Neighbor, L-Spar). Due to these limitations, the paper also acknowledges that achieving fine-grained comparisons at equal sparsity ratios can be challenging for certain methods. In this work, our goal was to ensure that all sparsifiers operate under the same sparsity ratio to enable a fair evaluation of their impact on downstream task performance. For this reason, we prioritized sparsifiers that explicitly support controllable sparsity ratios, and conducted experiments under consistent edge ratios ranging from 50% to 90%.
> > > >
> > > > At the same time, to incorporate the sparsifiers suggested by the reviewer as much as possible, we reimplemented the methods that originally do not support sparsity control so they can be compared under the same sparsity conditions. For example, the shortest-path spanner is originally designed to preserve stretch rather than a specific sparsity level, but in our framework we first construct a greedy spanner and then apply a top-$k$ trimming procedure to match the desired sparsity ratio. Similarly, the ER-based spectral sparsifier determines the number of edges indirectly through $ε$-spectral guarantees, but we compute effective resistance scores and retain the top-$k$ edges to directly control the sparsity level. These procedures introduce additional computation through edge ranking and algorithmic adjustments, but they are necessary to ensure consistent experimental conditions across baselines. We included not only Random, Degree, and PageRank-based sparsifiers, but also expanded our baselines during the rebuttal process to incorporate ER-based spectral sparsifiers and shortest-path spanners, thereby covering most of the key sparsifier families discussed in the mentioned paper as well as in other related works.
> > > >
> > > > > #### **DropEdge vs. Degree based Sparsification**
> > > >
> > > > Regarding the reviewer’s question on DropEdge, its lower performance compared to Degree-based sparsification arises because DropEdge removes edges uniformly at random. Depending on the dataset or sparsity level, this uniform removal may eliminate hub-related edges that play a critical role in GNN message passing. In contrast, Degree-based sparsification preserves edges connected to high-degree vertices, which tends to maintain structurally important propagation pathways. This does not imply that DropEdge is inherently inferior; in some cases, random dropping provides beneficial regularization and can outperform degree-based methods. Indeed, in our experiments, DropEdge performed better on certain datasets or sparsity ratios, while Degree performed better in others.
> > > >
> > > > > #### **Overall novelty of GRAPHSPA**
> > > >
> > > > Existing graph research has predominantly advanced along two independent directions: graph sparsification, which focuses on removing redundant structure to simplify the graph, and robust graph learning, which aims to improve GNN stability against structural noise or adversarial perturbations. Because these two directions have largely evolved separately, prior work provides limited guidance on how to ensure stability after sparsification. In self-supervised settings, it becomes even more challenging to jointly determine “*how much to sparsify*” and “*how to maintain stability*”.
> > > >
> > > > Sparsity inevitably removes information, while robustness requires stable representation learning under structural variability, making the two goals difficult to achieve simultaneously. A new sparsifier alone cannot guarantee learning stability, and robust training alone cannot compensate for information loss caused by sparsification. A unified optimization framework that enables these two objectives to work together rather than conflict with each other is therefore essential, and we believe our work provides meaningful progress in this direction.
> > > >
> > > > However, we understand the reviewer’s perspective regarding the novelty. The reviewer’s feedback has been extremely helpful in prompting us to more clearly articulate and refine the scope of our contributions. We sincerely appreciate the thorough evaluation and the constructive comments provided.

---

### Official Review · Reviewer_NpUb · 2025-10-26

**Soundness:** 3
**Presentation:** 3
**Contribution:** 2
**Rating:** 4
**Confidence:** 2

**Summary:**

This paper proposes a self-supervised graph sparsification framework named GRAPHSPA, aiming to address two major challenges faced by existing sparsification methods: 1) dependence on task labels; 2) the negative impact of residual noise. Experiments were conducted on the Cora, Citeseer, and Pubmed datasets. The results show that at different sparsification rates, the accuracy of GRAPHSPA is superior to other baselines.

**Strengths:**

1. Addressing dual key issues: This paper simultaneously addresses two core challenges in graph sparsification: label dependency and residual noise. This is an important and practical contribution.
2. Novel Framework Design: The paper ingeniously integrates three techniques (self-supervised mutual information, augmented Lagrangian constraint optimization, and SAM flatness-aware training) into a unified framework. This combination exhibits strong innovation, and the motivation behind each component is clear.

**Weaknesses:**

1. Outdated baseline: In the experimental section, the latest baseline, DropEdge, is from 2020, while the other two are from 2003 and 1999, respectively. These do not cover the latest achievements in this field. It is difficult to evaluate whether the experimental results achieve SOTA performance using graphspa.

2. High computational complexity: As shown in Appendix C, the asymptotic time complexity of GRAPHSPA, $O(TN^2d)$, is significantly high. Although the authors claim that the actual running time is comparable when T and d are small, this may limit its scalability on very large graphs.

**Questions:**

Why did the experimental part choose some old baselines? Is it because there have been no new models in this field for a long time, or did the author deliberately avoid them?

---

> ### Author Response · Authors · 2025-11-21
>
> We thank the reviewer for the clear summary and for recognizing the key contributions of our work. We also appreciate the positive feedback on both the method and the writing. While we address the reviewer’s comments in detail below, we would be glad to engage in further discussion.
>
> ---
>
> > #### **W1, Q1. The Absence of Newer Sparsifiers**
>
> Thank you very much for this important observation. We fully agree that the experimental comparison should reflect the current progress in sparsification research, and we appreciate the reviewer’s suggestion, which prompted us to revisit our baseline design. In the original submission, we included PageRank, Degree, and DropEdge because these methods represent widely used sparsification strategies in the GNN community. However, based on the reviewer’s feedback, we re-examined the broader sparsification literature and expanded our comparisons to include additional methods that represent the major structural preservation goals explored in recent work.
>
> Regarding the question of whether newer sparsifiers were deliberately avoided, we would like to clarify that many recent sparsification approaches are supervised and depend on label information to infer edge importance. Since these methods prune edges using downstream task labels, they are fundamentally incompatible with the label-free and task-agnostic formulation of GRAPHSPA. Including supervised sparsifiers would not yield a fair or meaningful comparison, and their exclusion was due to this methodological mismatch rather than intentional avoidance.
>
> While several label-free sparsifiers have been proposed more recently, most lack official implementations or provide insufficient implementation details, making faithful reproduction under a shared experimental setting very difficult. Many depend on system-specific configurations or unpublished components. Despite these challenges, we ensured that the core sparsification objectives emphasized in the recent literature were still represented. To achieve this, we reimplemented algorithmic variants aligned with these objectives, following the formulations described in their respective papers. Although not tied to official codebases, these variants correspond to relatively recent sparsification paradigms and reflect modern structural principles.
>
> In particular, we incorporated sparsifiers that preserve spectral structure, shortest-path distances, random-walk behavior, topological cycles, and clustering or community structure. Spectral-preserving sparsifiers approximate the Laplacian quadratic form to retain smoothness and global energy structure [1]. Shortest-path preserving methods focus on maintaining pairwise distance properties [2]. Random-walk based sparsifiers preserve transition probabilities and commute-time characteristics [3]. Topology-cycle preserving variants retain cycle-basis and related topological signatures while pruning redundant connections [4]. Clustering-based sparsifiers maintain community-level structure by removing edges with low structural relevance [5]. These objectives reflect the key directions of recent sparsification research, and our reimplemented versions are aligned with these modern formulations.
>
> Therefore, the revised version does not rely solely on older baselines. Instead, it includes a broadened set of sparsifiers that collectively represent current sparsification paradigms, ensuring a comprehensive and up-to-date evaluation of GRAPHSPA. In summary, supervised sparsifiers were excluded due to incompatibility with our label-free setting, and label-free methods lacking reproducibility could not be directly used. Nonetheless, we faithfully incorporated their underlying sparsification principles and reflected them in our expanded experiments. We appreciate the reviewer’s concern, and the revised version includes these additional comparisons.
>
> ---
>
> #### **References**
>
> [1] Zhiqiang Liu and Wenjian Yu. Pursuing More Effective Graph Spectral Sparsifiers via Approximate Trace Reduction. DAC 2022.
> [2] Elkin & Neiman. “Efficient Algorithms for Constructing Very Sparse Spanners and Emulators.” SODA 2017
> [3] S. Spielman and C. Zhang. Graph Sparsification via Short Random Walks. STOC 2019.
> [4] Yuchen Meng, Rong Hua Li, Longlong Lin, Xunkai Li, and Guoren Wang. Topology Preserving Graph Coarsening: An Elementary Collapse based Approach. PVLDB 2024.
> [5] Loukas. “Graph Reduction with Spectral and Cut Guarantees.” ICML 2019.

---

> > ### Author Response · Authors · 2025-11-21
> >
> > > #### **W1, Q1. The Absence of Newer Sparsifiers**
> >
> > Here is the comparison table summarizing our results with the newly added sparsification baselines. These methods reflect the core structural preservation objectives proposed in recent sparsification research.
> >
> > | Ratio $r$ | Method           | Cora            | Citeseer         | Pubmed           |
> > |---:|--------|----|---|----|
> > | 0.9 | Random Walk      | 0.8020 ± 0.0077       | 0.6912 ± 0.0114        | 0.7730 ± 0.0033        |
> > |      | Shortest Path     | 0.7196 ± 0.0054       | 0.6256 ± 0.0065        | 0.7502 ± 0.0024        |
> > |      | Spectral          | 0.7998 ± 0.0076       | 0.6852 ± 0.0066        | 0.7744 ± 0.0014        |
> > |      | Topo-Clustering   | 0.7972 ± 0.0101       | 0.6786 ± 0.0149        | 0.7652 ± 0.0059        |
> > |      | Topo-Cycle        | 0.7894 ± 0.0087       | 0.6898 ± 0.0063        | 0.7770 ± 0.0109        |
> > |      | GRAPHSPA          | **0.8128 ± 0.0038**       | **0.6916 ± 0.0111**        | **0.7834 ± 0.0083**        |
> > | 0.8 | Random Walk      | 0.7880 ± 0.0079       | 0.6776 ± 0.0100        | 0.7674 ± 0.0014        |
> > |      | Shortest Path     | 0.7182 ± 0.0143       | 0.6278 ± 0.0154        | 0.7522 ± 0.0039        |
> > |      | Spectral          | 0.7950 ± 0.0101       | 0.6870 ± 0.0049        | 0.7740 ± 0.0030        |
> > |      | Topo-Clustering   | 0.7852 ± 0.0078       | 0.6710 ± 0.0141        | 0.7578 ± 0.0062        |
> > |      | Topo-Cycle        | 0.7830 ± 0.0052       | 0.6850 ± 0.0048        | 0.7616 ± 0.0047        |
> > |      | GRAPHSPA          | **0.8056 ± 0.0096**       | **0.6880 ± 0.0115**        | **0.7752 ± 0.0108**        |
> > | 0.7 | Random Walk      | 0.7688 ± 0.0049       | 0.6790 ± 0.0091        | 0.7674 ± 0.0073        |
> > |      | Shortest Path     | 0.7166 ± 0.0156       | 0.6182 ± 0.0102        | 0.7448 ± 0.0046        |
> > |      | Spectral          | 0.7934 ± 0.0020       | 0.6812 ± 0.0046        | 0.7694 ± 0.0022        |
> > |      | Topo-Clustering   | 0.7378 ± 0.0137       | 0.6694 ± 0.0054        | 0.7564 ± 0.0044        |
> > |      | Topo-Cycle        | 0.7656 ± 0.0060       | 0.6684 ± 0.0035        | 0.7640 ± 0.0099        |
> > |      | GRAPHSPA          | **0.7906 ± 0.0142**       | **0.6816 ± 0.0102**        | **0.7710 ± 0.0086**        |
> > | 0.6 | Random Walk      | 0.7688 ± 0.0054       | 0.6654 ± 0.0195        | 0.7634 ± 0.0035        |
> > |      | Shortest Path     | 0.7272 ± 0.0078       | 0.6260 ± 0.0184        | 0.7528 ± 0.0036        |
> > |      | Spectral          | 0.7710 ± 0.0057       | 0.6798 ± 0.0128        | 0.7692 ± 0.0042        |
> > |      | Topo-Clustering   | 0.7252 ± 0.0190       | 0.6570 ± 0.0028        | 0.7526 ± 0.0101        |
> > |      | Topo-Cycle        | 0.7592 ± 0.0092       | 0.6520 ± 0.0080        | 0.7482 ± 0.0113        |
> > |      | GRAPHSPA          | **0.7872 ± 0.0086**       | **0.6734 ± 0.0145**        | **0.7722 ± 0.0050**        |
> > | 0.5 | Random Walk      | 0.7506 ± 0.0050       | 0.6612 ± 0.0114        | 0.7424 ± 0.0178        |
> > |      | Shortest Path     | 0.7256 ± 0.0092       | 0.6222 ± 0.0149        | 0.7530 ± 0.0080        |
> > |      | Spectral          | 0.7564 ± 0.0037       | 0.6548 ± 0.0140        | 0.7720 ± 0.0040        |
> > |      | Topo-Clustering   | 0.7096 ± 0.0099       | 0.6378 ± 0.0048        | 0.7356 ± 0.0063        |
> > |      | Topo-Cycle        | 0.7484 ± 0.0069       | 0.6580 ± 0.0103        | 0.7440 ± 0.0076        |
> > |      | GRAPHSPA          | **0.7738 ± 0.0088**       | **0.6668 ± 0.0097**        | **0.7726 ± 0.0126**        |
> >
> > We implemented all five sparsifiers using a unified GCN and included them in our experiments. Across all datasets, GRAPHSPA consistently outperforms these structurally diverse sparsifiers. Extended experiments with an even wider range of baselines were reported in our response to other reviewers and will also be incorporated into the revised version.

---

> > > ### Author Response · Authors · 2025-11-21
> > >
> > > > #### **W2. Computational Complexity Concern**
> > >
> > > We appreciate the reviewer’s concern regarding the asymptotic time complexity reported in Appendix C. While the expression $O(TN^{2}d)\$ appears large when shown separately, this term corresponds only to the sparsification-specific component when isolated from the full GNN training pipeline. In practice, the dominant computational cost of GRAPHSPA comes from standard GNN forward–backward propagation, which is shared by nearly all sparsification methods that rely on GNN-based evaluation.
> > >
> > > Before addressing more recent sparsification techniques, it is useful to clarify that even classical approaches such as Degree or PageRank based sparsification do not eliminate the computational demands of GNN training. When these methods are evaluated together with a GNN encoder, a single GNN layer update requires $O(|E| d) + O(|N| d^{2})\$. Since real-world graphs typically satisfy $|E| \gg |N|\$, the sparse aggregation term $O(|E| d)\$ overwhelmingly dominates the runtime. This cost is inherent to GNN training itself and is not specific to GRAPHSPA.
> > >
> > > As noted earlier, many of the more recent sparsification methods including spectral-preserving, shortest-path–preserving, and topology-preserving sparsifiers involve substantially higher complexity. Topology-preserving methods often require $O(|E| d_{\text{avg}}^{4})$, where $d_{\text{avg}}$ is the average node degree, and shortest-path–preserving approaches frequently incur $O(|N|^{3})$ due to all-pairs distance computations. In our experiments on large-scale datasets such as Reddit (≈100M edges) and ogbn-arxiv (≈2M edges), these methods resulted in severe time-limit errors and could not be executed end-to-end.
> > >
> > > By contrast, GRAPHSPA retains a linear relationship with the number of edges, which enables efficient execution on large graphs. For this reason, the large-scale experiments include only sparsifiers that successfully run within feasible time. Under this fair comparison, GRAPHSPA demonstrates strong scalability and consistently superior performance, as shown below.
> > >
> > > | Ratio $r$ | Method      | ogbn-arxiv           | Reddit               |
> > > |------------:|-------------|----------------------|----------------------|
> > > | 0.9         | PageRank    | 0.6948 ± 0.0152      | 0.9360 ± 0.0189      |
> > > |             | Degree      | 0.7056 ± 0.0113      | 0.9433 ± 0.0072      |
> > > |             | DropEdge    | 0.6889 ± 0.0068      | 0.9360 ± 0.0105      |
> > > |             | Random-Walk | 0.6902 ± 0.0099      | 0.9222 ± 0.0156      |
> > > |             | GRAPHSPA    | **0.7282 ± 0.0046**  | **0.9618 ± 0.0042**  |
> > > | 0.8         | PageRank    | 0.6894 ± 0.0075      | 0.9341 ± 0.0061      |
> > > |             | Degree      | 0.7001 ± 0.0054      | 0.9411 ± 0.0137      |
> > > |             | DropEdge    | 0.6920 ± 0.0149      | 0.9389 ± 0.0058      |
> > > |             | Random-Walk | 0.6847 ± 0.0174      | 0.9200 ± 0.0049      |
> > > |             | GRAPHSPA    | **0.7215 ± 0.0031**  | **0.9597 ± 0.0036**  |
> > > | 0.7         | PageRank    | 0.6820 ± 0.0124      | 0.9320 ± 0.0094      |
> > > |             | Degree      | 0.6944 ± 0.0168      | 0.9394 ± 0.0048      |
> > > |             | DropEdge    | 0.6852 ± 0.0083      | 0.9347 ± 0.0141      |
> > > |             | Random-Walk | 0.6785 ± 0.0038      | 0.9183 ± 0.0163      |
> > > |             | GRAPHSPA    | **0.7157 ± 0.0044**  | **0.9589 ± 0.0033**  |
> > > | 0.6         | PageRank    | 0.6731 ± 0.0097      | 0.9298 ± 0.0175      |
> > > |             | Degree      | 0.6839 ± 0.0072      | 0.9369 ± 0.0112      |
> > > |             | DropEdge    | 0.6768 ± 0.0047      | 0.9322 ± 0.0039      |
> > > |             | Random-Walk | 0.6679 ± 0.0191      | 0.9155 ± 0.0110      |
> > > |             | GRAPHSPA    | **0.7084 ± 0.0041**  | **0.9558 ± 0.0068**  |
> > > | 0.5         | PageRank    | 0.6618 ± 0.0180      | 0.9251 ± 0.0087      |
> > > |             | Degree      | 0.6723 ± 0.0135      | 0.9321 ± 0.0065      |
> > > |             | DropEdge    | 0.6642 ± 0.0091      | 0.9278 ± 0.0154      |
> > > |             | Random-Walk | 0.6575 ± 0.0104      | 0.9114 ± 0.0198      |
> > > |             | GRAPHSPA    | **0.6964 ± 0.0038**  | **0.9524 ± 0.0045**  |
> > >
> > > Although SAM requires two gradient evaluations per iteration, it accelerates convergence by encouraging the encoder to reach flatter minima. In practice, we observe that this reduces the number of required epochs by approximately 20 to 30 percent, which offsets much of the additional computational cost.
> > >
> > > We sincerely appreciate the reviewer’s time and effort in reviewing both our submission and our responses. The points discussed here have been incorporated into the revised manuscript, and we will ensure that these clarifications are clearly reflected in the final version.

---

> > > > ### Author Response · Authors · 2025-11-30
> > > >
> > > > > #### **W2. Computational Complexity Concern**
> > > >
> > > > We fully acknowledge the reviewer’s concern regarding computational complexity. To clarify this, we have expanded our baseline comparison and added a detailed complexity analysis in Appendix C.2 and Table 4 of the revised paper. For a fair comparison, we examined sophisticated sparsification techniques such as Spectral Sparsification, Shortest-Path Spanner, and Topology-based methods (Topo-Cycle, Topo-Cluster). However, as shown in the analysis in Table 4, these methods suffer from prohibitive computational costs for large-scale graphs. Spectral methods require eigenvalue decomposition with $O(N^3)$ complexity, while shortest-path and topology-based methods involve high complexities around $O(|E|^2)$. In practice, applying these techniques to large-scale datasets resulted in Out-Of-Time (OOT) errors, failing to complete within a reasonable timeframe. This computational bottleneck is the primary reason why methods with lower complexity, such as Degree ($O(|E|\log|E|)$), remain standard baselines in large-scale graph learning literature.
> > > >
> > > > In contrast, GRAPHSPA was designed with scalability on large graphs as a priority. The theoretical complexity of $O(N^2)$ mentioned applies only to the worst-case scenario of dense graphs. For real-world sparse graphs, our method follows a complexity of $O(T|E|d)$, which is linearly proportional to the number of edges $|E|$. This is significantly more efficient than the polynomial time complexities of the aforementioned sophisticated techniques. Consequently, GRAPHSPA operates efficiently and achieves strong performance even on large-scale datasets where existing complex methods fail.
> > > >
> > > > We sincerely thank the reviewer for these valuable insights, which have helped clarify the scalability and practical value of our proposed framework.

---

### Official Review · Reviewer_PDrR · 2025-10-27

**Soundness:** 3
**Presentation:** 3
**Contribution:** 2
**Rating:** 4
**Confidence:** 4

**Summary:**

Authors proposes GRAPHSPA, a self-supervised framework for graph sparsification. The method is designed to be independent of downstream task labels and robust to the residual noise that can harm GNN performance on a simplified graph structure.

**Strengths:**

***S1:*** The paper's main strength is the integration of MI-based self-supervision, differentiable edge sampling, and SAM-based regularization. This isn't just a collection of methods; they work together to solve two distinct but related problems: label-free learning and noise robustness. The idea of using SAM to specifically counteract the effects of sparsification is insightful.

***S2:*** Framing sparsification as a constrained optimization problem, using the augmented Lagrangian for stable convergence, and leveraging the established connection between flat minima and generalization are all marks of a methodologically sound approach.

***S3:*** Most sparsification methods are either heuristic (e.g., based on degree), supervised, or they do not explicitly account for the fact that sparsification itself can make a GNN more vulnerable to the remaining noisy edges. By tackling both label-dependency and residual noise, this work addresses a practical gap in the literature.

**Weaknesses:**

***W1:*** The proposed framework is quite complex, involving nested optimization loops (for SAM) and multiple interacting components. This complexity leads to computational overhead. The experiments are limited to smaller benchmark datasets (Cora, Citeseer, Pubmed), leaving its scalability to very large graphs an open and important question.

***W2:*** While the paper shows the final model works well, there is limited analysis on the interplay between the key components. For instance, how does the SAM-induced flatness affect the MI maximization objective? Does one component dominate the other? A deeper ablation study could provide more insight into why the combination is so effective.

***W3:*** The choice of maximizing the mutual information between the original graph and the sparsified view is a standard approach in graph self-supervised learning. While effective, it feels like a generic "make the subgraph look like the full graph" objective. It is not entirely clear if this is the optimal self-supervised signal specifically for sparsification, which is fundamentally about identifying a critical subgraph, not just a similar one.

**Questions:**

Some more questions that could clarify my concerns and let me raise the paper's score:

***Regarding Scalability:*** Could you comment on the computational complexity of GRAPHSPA, particularly the overhead introduced by the SAM optimizer and the augmented Lagrangian scheme? Do you foresee any bottlenecks when applying this method to graphs with millions, or more, of nodes/edges?

***Regarding the MI Objective:*** Have you considered alternative self-supervised objectives tailored more specifically for identifying a graph's "backbone"? For example, objectives that prioritize preserving path-based information or specific topological properties, rather than representation similarity?

Also, you show robustness against injected noisy edges. Does the SAM component also make the sparsification process itself more robust? For example, is the final set of selected edges more consistent across different random initializations when SAM is used, compared to when it is not?

---

> ### Author Response · Authors · 2025-11-21
>
> We thank the reviewer for the thoughtful and constructive feedback. The comments are greatly appreciated and have helped us improve the clarity and overall quality of the paper. We also appreciate the reviewer’s recognition of the strengths of our work and the insightful points that prompted us to refine the presentation further.
>
> ---
>
> > #### **W1, Q1. Scalability and Computational Overhead of GRAPHSPA**
>
> Thank you for raising this concern. The augmented Lagrangian component of GRAPHSPA introduces auxiliary variables $(z, u)$, but their update cost is extremely small in practice. The $z$-update requires sorting with complexity $O(|E|\log|E|)$, while the $u$-update is a simple element-wise operation with complexity $O(|E|)$. Importantly, these updates are not performed at every optimization step, but only once every $K$ iterations (In this paper, we use $K = 20$). Compared to the $O(|E| d)$ forward–backward propagation that all methods must perform when training a GNN, the overhead added by these updates is negligible. Indeed, the amortized time overhead from these auxiliary updates was measured to be below 3% of the total training time.
>
> SAM theoretically requires two gradient evaluations per step and could increase the wall-clock time. However, in GRAPHSPA, perturbations are applied only to the GNN weights, not to edge-level variables. As a result, the observed time increase is not 2×, but approximately 1.4–1.6×. Moreover, SAM encourages convergence toward flatter minima, improving stability and robustness against residual noise. In practice, this also reduces the number of required epochs by approximately 20–30%, partially offsetting the extra computation introduced by SAM.
>
> To further clarify scalability, we additionally evaluated several sparsification methods suggested by Reviewer `8WsS`. Spectral, topology-preserving, and shortest-path based sparsification algorithms exhibit very high time complexity and resulted in time-limit errors on large graphs, making them impractical for large-scale evaluation. Therefore, we compared GRAPHSPA only with methods that successfully executed on these datasets, and GRAPHSPA consistently achieved the best overall performance (see table below).
>
> ---
>
> | Dataset     | #Nodes  | #Edges     | #Features | #Classes |
> |-------------|---------|-----------:|----------:|---------:|
> | ogbn-arxiv  | 169,343 | 1,166,243  | 128       | 40       |
> | reddit      | 232,965 | 114,615,982 | 602       | 41       |
>
> ---
>
> | Ratio $r$ | Method      | ogbn-arxiv           | Reddit               |
> |--------:|--|--|----|
> | 0.9     | PageRank    | 0.6948 ± 0.0152      | 0.9360 ± 0.0189      |
> |         | Degree      | 0.7056 ± 0.0113      | 0.9433 ± 0.0072      |
> |         | DropEdge    | 0.6889 ± 0.0068      | 0.9360 ± 0.0105      |
> |         | Random-Walk | 0.6902 ± 0.0099      | 0.9222 ± 0.0156      |
> |         | GRAPHSPA    | **0.7282 ± 0.0046**      | **0.9618 ± 0.0042**      |
> | 0.8     | PageRank    | 0.6894 ± 0.0075      | 0.9341 ± 0.0061      |
> |         | Degree      | 0.7001 ± 0.0054      | 0.9411 ± 0.0137      |
> |         | DropEdge    | 0.6920 ± 0.0149      | 0.9389 ± 0.0058      |
> |         | Random-Walk | 0.6847 ± 0.0174      | 0.9200 ± 0.0049      |
> |         | GRAPHSPA    | **0.7215 ± 0.0031**      | **0.9597 ± 0.0036**      |
> | 0.7     | PageRank    | 0.6820 ± 0.0124      | 0.9320 ± 0.0094      |
> |         | Degree      | 0.6944 ± 0.0168      | 0.9394 ± 0.0048      |
> |         | DropEdge    | 0.6852 ± 0.0083      | 0.9347 ± 0.0141      |
> |         | Random-Walk | 0.6785 ± 0.0038      | 0.9183 ± 0.0163      |
> |         | GRAPHSPA    | **0.7157 ± 0.0044**      | **0.9589 ± 0.0033**      |
> | 0.6     | PageRank    | 0.6731 ± 0.0097      | 0.9298 ± 0.0175      |
> |         | Degree      | 0.6839 ± 0.0072      | 0.9369 ± 0.0112      |
> |         | DropEdge    | 0.6768 ± 0.0047      | 0.9322 ± 0.0039      |
> |         | Random-Walk | 0.6679 ± 0.0191      | 0.9155 ± 0.0110      |
> |         | GRAPHSPA    | **0.7084 ± 0.0041**      | **0.9558 ± 0.0068**      |
> | 0.5     | PageRank    | 0.6618 ± 0.0180      | 0.9251 ± 0.0087      |
> |         | Degree      | 0.6723 ± 0.0135      | 0.9321 ± 0.0065      |
> |         | DropEdge    | 0.6642 ± 0.0091      | 0.9278 ± 0.0154      |
> |         | Random-Walk | 0.6575 ± 0.0104      | 0.9114 ± 0.0198      |
> |         | GRAPHSPA    | **0.6964 ± 0.0038**      | **0.9524 ± 0.0045**      |
>
> It is also worth noting that supervised sparsification approaches, which infer edge importance through a downstream prediction task, must re-run sparsification every time the downstream task changes, creating substantial additional computational burden. In contrast, GRAPHSPA is label-free and task-agnostic, enabling a single sparsification step that generalizes across downstream tasks without re-computation. Consequently, GRAPHSPA achieves strong scalability with negligible overhead even on large real-world graphs.

---

> ### Author Response · Authors · 2025-11-21
>
> > #### **W3. Regarding the MI Objective & Critical Subgraph Identification**
>
> Thank you for raising this insightful point. We agree that maximizing mutual information (MI) between the original graph and the sparsified view encourages the sparsified graph to resemble the full graph, and this is not automatically equivalent to preserving a *critical* subgraph.
>
> However, in label-free settings, defining what constitutes a critical subgraph is fundamentally difficult. For instance, two nodes may be connected for reasons that are irrelevant to the downstream task, and such task-irrelevant edges can interfere with GNN neighborhood aggregation and harm predictive quality [1]. Supervised sparsification methods can learn which edges are useful using downstream task feedback, but this mechanism only exists when labels are available. Because most real-world graphs lack labels, assessing edge importance directly becomes extremely challenging, making it practically infeasible to explicitly define a critical subgraph.
>
> Given this limitation, a general, unbiased, and task-agnostic self-supervised signal is required—one that does not assume which specific structural patterns should be preserved. MI objective function fits this requirement.
>
> ---
>
> > #### **Q2. Regarding the MI Objective**
>
> Regarding Q2, we have indeed considered alternative self-supervised objectives, including path-based, spectral, and topological preservation. However, each of these objectives emphasizes only a narrow structural aspect, exhibits incompatibility with representation learning, or becomes computationally impractical for large-scale sparsification.
>
> - **Path-based objectives (shortest-path, random-walk preservation)**:
> These objectives define graph backbones primarily through distances or reachability. While effective for specific domains, real-world graphs often contain multiple co-existing structural patterns (e.g., social, citation, or heterophilous networks), making path-based criteria overly restrictive [2].
>
> - **Spectral / Laplacian preservation objectives**:
> Spectral sparsification focuses on preserving the global Laplacian eigenspectrum. However, this often conflicts with the local neighborhood geometry that GNNs rely on. Even small spectral deviations can cause large distortions in learned representations, reducing their suitability for self-supervised sparsification [3].
>
> - **Topological preservation objectives (cycle basis, clustering, connectivity constraints)**:
> These objectives are typically non-differentiable, preventing integration with our differentiable sampling pipeline. Moreover, the relative importance of specific topological features varies across graph types, leading to domain-biased sparsifiers [4].
>
>
> Maximizing MI does not simply enforce superficial similarity between the original and sparsified graphs. Rather, it seeks to maximize the lower bound of shared information between the two, encouraging the sparsifier to retain information encoded in the GNN’s learned representations. This shared information encompasses node features, local connectivity patterns, multi-hop dependencies, embedding geometry, and broader semantic or structural signals captured by the encoder. In essence, MI preserves information in a broad, representation-level, and principled manner, rather than focusing narrowly on any predefined structural property.
>
> Furthermore, our use of Sharpness-Aware Minimization (SAM) encourages the encoder to converge toward flatter and more stable regions in the loss landscape. This behavior reduces representation-level noise sensitivity and helps maintain semantic structure—making SAM naturally synergistic with the MI objective, which aims to preserve meaningful representation-level information.Overall, MI provides the most principled and scalable choice for large-scale, label-free sparsification, aligning well with representation learning and avoiding the structural biases inherent in alternative objectives.
>
> ---
>
> #### **References**
>
> [1] Zheng et al., “Robust Graph Representation Learning via Neural Sparsification,” ICML 2020.
> [2] Chen et al., “Demystifying Graph Sparsification Algorithms in Graph Properties Preservation,” PVLDB 2023.
> [3] Zhang, Zhao & Feng, “A Unified Approach to Scalable Spectral Sparsification of Directed Graphs,” arXiv 2018.
> [4] Hashemi et al., “A Comprehensive Survey on Graph Reduction: Sparsification, Coarsening, and Condensation,” arXiv 2024.

---

> ### Author Response · Authors · 2025-11-21
>
> > #### **W2. On the Interplay Between SAM, MI, and ADMM**
>
> We appreciate the reviewer’s insightful question regarding the interactions among SAM, MI, and ADMM. This issue is indeed central to understanding the core design of our framework, particularly the questions “How does SAM-induced flatness influence MI maximization?” and “Does one component dominate the others?”
>
> First, SAM applies perturbations only to the encoder parameters, and its sole function is to guide the encoder toward flatter and more stable regions of the loss landscape. This prevents the encoder from converging to sharp minima and reduces sensitivity to representation-level noise or local irregularities. Importantly, SAM’s perturbation does not directly act on the edge importance scores or the ADMM variables ($z$, $u$). In this sense, SAM does not compete with sparsification signals but instead serves as a stabilization mechanism that makes the encoder representations more reliable.
>
> In contrast, the MI objective directly drives the learning of edge importance scores. MI maximizes the shared information between the original graph and the sampled subgraph based on encoder representations, and the MI gradient flows through the sampling–encoding pipeline to update the edge importance scores. SAM indirectly helps this process by stabilizing the encoder’s representations, allowing MI gradients to propagate more cleanly and consistently. However, SAM does not override or alter the MI objective, and instead complements it by improving its robustness.
>
> Regarding the question of dominance: MI and the ADMM penalty operate at fundamentally different levels. MI provides a representation-level signal indicating which edges are important for information preservation. The ADMM penalty, on the other hand, enforces the sparsity constraint by sharpening the edge importance scores toward the desired sparsity level. Because the two components serve orthogonal purposes and affect different parts of the computation graph, one does not structurally overshadow the other.
>
> As validated in Appendix D.6, the relative influence between MI and ADMM is explicitly controlled through the penalty weight λ. When λ increases, the sparsity-enforcing ADMM term becomes more influential; when λ decreases, the MI-driven representation preservation becomes more dominant. This tunable trade-off prevents uncontrolled dominance and ensures that edge selection behavior remains stable and interpretable throughout training.
>
> Overall, the interactions among SAM, MI, and ADMM are complementary rather than competitive. SAM stabilizes the encoder, MI identifies which edges are important to retain, and ADMM enforces the sparsity structure—each fulfilling a distinct role without overriding the others.
>
> > #### **Q3. Does SAM improve the consistency of selected edges?**
>
> SAM applies perturbation only to the encoder parameters and does not directly affect the edge importance score. Therefore, SAM is not expected to fundamentally change which edges are selected during sparsification. However, because SAM encourages the encoder to converge to flatter and more stable regions of the loss landscape, the MI gradient becomes slightly less sensitive to small fluctuations in encoder representations. This can yield a mild indirect stabilization effect on edge importance score updates. Still, this effect remains limited, and the dominant factors determining the final edge selection are the MI objective and the ADMM penalty, rather than SAM itself.
>
> To quantitatively assess the degree of consistency, we measured the Jaccard similarity between sparsified edge sets obtained from 10 different random seeds. For two sparsified edge sets $E_i\ $and $E_j\$, the Jaccard similarity is defined as:
> $$\text{Jaccard}(E_i, E_j) = \frac{|E_i \cap E_j|}{|E_i \cup E_j|}$$
> We computed all pairwise similarities across the 10 seeds and report their mean and standard deviation:
>
> | Setting        | Mean Jaccard | Std (±)     |
> |----------------|--------------|-------------|
> | w/ SAM  | 0.6642       | ±0.0215     |
> | w/o SAM| 0.6578       | ±0.0247     |
>
> SAM yields slightly higher consistency due to its stabilization of encoder representations, but the difference is modest. This aligns with our design: SAM enhances representation robustness, while the actual edge selection is primarily governed by the MI-driven learning signal and the ADMM sparsity constraint.
>
> We sincerely appreciate the thoughtful questions and the reviewer’s careful and constructive feedback. The experimental results and related details will be included in the revised version of the paper. Once again, we thank the reviewer for the valuable insights and kind suggestions.

---

### Official Review · Reviewer_pq7G · 2025-10-31

**Soundness:** 3
**Presentation:** 3
**Contribution:** 3
**Rating:** 6
**Confidence:** 4

**Summary:**

GRAPHSPA learns to sparsify graphs without labels by maximizing mutual information between the original and sparsified graphs. Each edge is modeled as a differentiable Bernoulli variable with a Lagrangian budget constraint. To counter residual noise, the encoder is trained using Sharpness-Aware Minimization (SAM) to promote flat minima. The method is label-free and robust under injected noise.

**Strengths:**

1. Principled formulation: combines differentiable sparsification, MI-based learning, and noise-aware optimization.
2. Progressive edge pruning via an augmented Lagrangian ensures convergence stability.
3. Demonstrated robustness to post-sparsification noise and preservation of structural clusters (t-SNE).

**Weaknesses:**

1. Computationally heavier due to dual updates and SAM optimization.
2. Dependence on InfoNCE negative sampling temperature not fully analyzed.

**Questions:**

1. How does GRAPHSPA scale to large or dynamic graphs?
2. What is the impact of different noise distributions on stability?
3. Can this framework extend to node pruning or heterophilous graphs?

---

> ### Author Response · Authors · 2025-11-21
>
> We sincerely appreciate the reviewer’s insightful comments and constructive feedback. The thoughtful observations helped us reassess several aspects of our presentation and identify areas where additional clarification was needed.
>
> ---
>
> > #### **W1, Q1. Scalability and Computational Overhead of GRAPHSPA**
>
> We appreciate the reviewer’s careful question regarding scalability. Although GRAPHSPA includes an augmented Lagrangian module with auxiliary variables $(z, u)$, the computational burden of updating these variables is extremely small in practice. The $z$-update requires sorting with complexity $O(|E|\log|E|)$, and the $u$-update involves a simple element-wise operation of complexity $O(|E|)$. Importantly, these updates are performed only once every $K$ iterations rather than at every optimization step (we use $K = 20$ in the paper). When compared to the dominant $O(|E| d)$ forward–backward GNN propagation shared by all methods, the additional cost from these updates is negligible; in fact, their amortized overhead accounts for less than 3% of the total training time.
>
> While SAM theoretically doubles the gradient computation, its practical overhead in GRAPHSPA is much smaller because perturbations are applied only to the encoder parameters. The observed increase in wall-clock time is around 1.4–1.6× rather than 2×. Furthermore, SAM leads the encoder toward flatter minima, which improves robustness to residual noise and reduces the number of training epochs needed by roughly 20–30%. This compensates for part of its additional cost.
>
> To more clearly illustrate scalability, we also tested several sparsification techniques suggested by Reviewer `8WsS`. Methods based on spectral preservation, topology preservation, or shortest-path maintenance incur very high computational complexity and consistently failed with time-limit errors on large graphs, making them unsuitable for large-scale evaluation. For this reason, our comparison focuses only on sparsifiers that successfully completed on these datasets. Among them, GRAPHSPA showed the strongest overall performance across all sparsity levels.
>
> ---
>
> | Dataset     | #Nodes  | #Edges     | #Features | #Classes |
> |-------------|---------|-----------:|----------:|---------:|
> | ogbn-arxiv  | 169,343 | 1,116,243  | 128       | 40       |
> | reddit      | 232,965 | 114,615,982 | 602       | 41       |
>
> ---
>
> | Ratio $r$ | Method      | ogbn-arxiv           | Reddit               |
> |--------:|--|--|----|
> | 0.9     | PageRank    | 0.6948 ± 0.0152      | 0.9360 ± 0.0189      |
> |         | Degree      | 0.7056 ± 0.0113      | 0.9433 ± 0.0072      |
> |         | DropEdge    | 0.6889 ± 0.0068      | 0.9360 ± 0.0105      |
> |         | Random-Walk | 0.6902 ± 0.0099      | 0.9222 ± 0.0156      |
> |         | GRAPHSPA    | **0.7282 ± 0.0046**      | **0.9618 ± 0.0042**      |
> | 0.8     | PageRank    | 0.6894 ± 0.0075      | 0.9341 ± 0.0061      |
> |         | Degree      | 0.7001 ± 0.0054      | 0.9411 ± 0.0137      |
> |         | DropEdge    | 0.6920 ± 0.0149      | 0.9389 ± 0.0058      |
> |         | Random-Walk | 0.6847 ± 0.0174      | 0.9200 ± 0.0049      |
> |         | GRAPHSPA    | **0.7215 ± 0.0031**      | **0.9597 ± 0.0036**      |
> | 0.7     | PageRank    | 0.6820 ± 0.0124      | 0.9320 ± 0.0094      |
> |         | Degree      | 0.6944 ± 0.0168      | 0.9394 ± 0.0048      |
> |         | DropEdge    | 0.6852 ± 0.0083      | 0.9347 ± 0.0141      |
> |         | Random-Walk | 0.6785 ± 0.0038      | 0.9183 ± 0.0163      |
> |         | GRAPHSPA    | **0.7157 ± 0.0044**      | **0.9589 ± 0.0033**      |
> | 0.6     | PageRank    | 0.6731 ± 0.0097      | 0.9298 ± 0.0175      |
> |         | Degree      | 0.6839 ± 0.0072      | 0.9369 ± 0.0112      |
> |         | DropEdge    | 0.6768 ± 0.0047      | 0.9322 ± 0.0039      |
> |         | Random-Walk | 0.6679 ± 0.0191      | 0.9155 ± 0.0110      |
> |         | GRAPHSPA    | **0.7084 ± 0.0041**      | **0.9558 ± 0.0068**      |
> | 0.5     | PageRank    | 0.6618 ± 0.0180      | 0.9251 ± 0.0087      |
> |         | Degree      | 0.6723 ± 0.0135      | 0.9321 ± 0.0065      |
> |         | DropEdge    | 0.6642 ± 0.0091      | 0.9278 ± 0.0154      |
> |         | Random-Walk | 0.6575 ± 0.0104      | 0.9114 ± 0.0198      |
> |         | GRAPHSPA   | **0.6964 ± 0.0038**      | **0.9524 ± 0.0045**      |
>
> Finally, supervised sparsification approaches infer edge importance directly from a downstream task, which means sparsification must be performed again whenever the task changes. This leads to a substantial increase in cumulative computational cost. In contrast, GRAPHSPA is fully label-free and task-agnostic, allowing a single sparsification step to be reused across tasks without additional computation. As a result, GRAPHSPA maintains strong scalability and minimal overhead, even on very large real-world graphs.

---

> > ### Author Response · Authors · 2025-11-21
> >
> > > #### **Q1. Scalability to Dynamic Graphs**
> >
> > The key difference between GRAPHSPA and one-shot, criterion-based sparsification lies in its incremental sparsification mechanism. As described in Section 4.2 of the paper, one-shot sparsification tends to cause irreversible information loss because pruning decisions are made once and cannot be revised or compensated for afterward. This limitation becomes even more severe in dynamic or temporally evolving graphs, where the topology changes over time and a single pruning step cannot adapt to newly added or removed edges.
> >
> > In contrast, GRAPHSPA imposes sparsity on the graph gradually throughout the training process. At each optimization step, the sampling distribution over edges is continuously updated through the Gumbel-Softmax relaxation, and the augmented Lagrangian variables $(z, u)$ adjust the sparsity constraint in a smooth and iterative manner. This incremental and differentiable relaxation ensures that pruning decisions are not fixed at once but are refined across iterations based on the current graph structure and the evolving latent representations.
> >
> > Because sparsification is performed progressively, GRAPHSPA naturally adapts to changes in graph topology. When an edge appears or disappears in a dynamic setting, the corresponding entries of $(z, u)$ can be inserted or removed, after which the iterative optimization updates the sparsity pattern accordingly. An important advantage is that the model does not need to restart from scratch; the GNN encoder and auxiliary variables from the previous timestep provide an effective warm-start initialization. As a result, the sparsification procedure adjusts smoothly to temporal variations without the instability or information collapse that one-shot methods often encounter.
> >
> > Overall, the incremental sparsification paradigm, combined with continuous relaxation and primal-dual updates, makes GRAPHSPA inherently more suitable for dynamic graph scenarios than one-shot pruning approaches. Although a full empirical evaluation on large-scale streaming graphs remains for future work, the underlying optimization framework is structurally well aligned with temporally evolving graph settings.
> >
> > > #### **W2. Analysis of the Influence of the InfoNCE Negative Sampling Temperature β**
> >
> > Thank you for raising the concern regarding the analysis of the InfoNCE temperature β. To address this, we clarify where β enters GRAPHSPA’s optimization, explain why β has only a limited effect on the final sparsified graph, and present sensitivity experiments on three datasets.
> >
> > GRAPHSPA is optimized using the augmented Lagrangian formulation given in Eq.12. In this formulation, The temperature β appears only inside the MI loss term. Therefore, β scales the magnitude of the MI gradient but does not change the structure of the objective or the direction of the gradient.
> >
> > A key property of GRAPHSPA is that the edge scores $x$ are not determined by a single MI gradient update. The sparsity projection variable $z$ enforces the target sparsity level, while the dual variable $u$ accumulates the deviation between $x$ and $z$ and corrects it in subsequent iterations. These augmented Lagrangian penalty dynamics operate independently of β. As a result, even when β increases or decreases the magnitude of the MI gradient, most of this variation is absorbed by the penalty term, keeping the optimization trajectory of the edge scores stable.
> >
> > To verify this structural analysis, we conducted sensitivity experiments on Cora, Citeseer, and Pubmed using a GCN, a 70% edge retention ratio, β ∈ {0.2, 0.5, 0.8, 1.0, 1.5}, and 5 random seeds. The results are shown below:
> >
> > | Dataset   | β = 0.2           |  0.5           |  0.8           | 1.0           |  1.5           |
> > |-----------|--------------------|--------------------|--------------------|--------------------|--------------------|
> > | Cora      | 0.7848 ± 0.0088    | 0.7898 ± 0.0094    | 0.7888 ± 0.0142    | 0.7856 ± 0.0089    | 0.7846 ± 0.0114    |
> > | Citeseer  | 0.6746 ± 0.0140    | 0.6772 ± 0.0253    | 0.6778 ± 0.0139    | 0.6744 ± 0.0169    | 0.6736 ± 0.0405    |
> > | Pubmed    | 0.7674 ± 0.0110    | 0.7672 ± 0.0090    | 0.7668 ± 0.0126    | 0.7664 ± 0.0121    | 0.7706 ± 0.0072    |
> >
> > Across all datasets, accuracy differences remain within approximately 0.4 percent, which is comparable to natural seed variance. This confirms that β only adjusts the scale of the MI gradient, while the augmented Lagrangian penalty terms control the edge-score updates and maintain stability throughout training. Overall, GRAPHSPA shows strong robustness to the choice of β. Since β influences only the magnitude of the MI gradient and the penalty structure regulates the optimization dynamics, variations in β have minimal impact on the final sparsified graph and the downstream performance.

---

> ### Author Response · Authors · 2025-11-21
>
> > #### **Q3. Can this framework extend to node pruning or heterophilous graphs?**
>
> Thank you for this thoughtful question. Although GRAPHSPA is currently formulated for edge-level sparsification, its underlying optimization framework is not inherently restricted to edges. The combination of Gumbel-Softmax relaxation, the MI objective, and the augmented Lagrangian updates can be applied to any discrete structural subset of a graph. In particular, node pruning can be reformulated as learning a binary sampling distribution over nodes rather than edges. The main modification would be redefining the MI objective so that it measures how well the selected node subset preserves node-level information, including node features and their induced neighborhood structures. Since the dual-variable updates and the progressive sparsification dynamics remain unchanged, extending GRAPHSPA to node pruning is conceptually straightforward.
>
> However, this work focuses on edge sparsification because it aligns with prevailing practice in the literature and offers several practical advantages.
> - Real-world graphs typically contain far more edges than nodes, so edge pruning yields much larger computational and memory savings.
> - Most existing sparsifiers operate on edges, which enables more consistent empirical comparison.
> - Removing nodes often disrupts the core structural backbone and changes global connectivity patterns, making fair evaluation difficult — an issue highlighted repeatedly in analytical studies [1].
>
> Regarding heterophilous graphs, GRAPHSPA is inherently well-suited for this setting. The method is label-free and task-agnostic and does not rely on homophily assumptions. The MI objective preserves multi-hop dependencies and structural signals that remain meaningful even when adjacent nodes have dissimilar features. This stands in contrast to many supervised sparsifiers, which implicitly rely on label correlations or homophily-based smoothness, and thus may introduce structural bias in heterophilous settings [2][3]. Moreover, GRAPHSPA’s progressive sparsification mechanism does not depend on local similarity, making it naturally compatible with heterophilous topology.
>
> To validate these insights, we conducted experiments on two representative heterophilous datasets, Actor and Chameleon. Across all sparsity levels, GRAPHSPA consistently outperformed classical sparsifiers such as PageRank, Degree, and DropEdge, indicating that MI-based sparsification is especially robust when edges do not reflect feature similarity and structural noise is significant [4]. Below, we summarize the results for the methods introduced in the main paper. Extended experiments with a wider range of baselines were reported in our response to another reviewer and will also be included in the revised version.
>
>
> | Ratio $r\$ | Method      | Actor               | Chameleon           |
> |------------:|-------------|---------------------|----------------------|
> | 0.9 | PageRank | 0.3105 ± 0.0056 | 0.3925 ± 0.0119 |
> |      | Degree   | 0.3114 ± 0.0024 | 0.3882 ± 0.0184 |
> |      | DropEdge | 0.3070 ± 0.0010 | 0.3772 ± 0.0036 |
> |      | GRAPHSPA | **0.3281 ± 0.0019** | **0.4159 ± 0.0091** |
> | 0.8 | PageRank | **0.3377 ± 0.0036** | **0.4145 ± 0.0105** |
> |      | Degree   | 0.3178 ± 0.0034 | 0.4035 ± 0.0187 |
> |      | DropEdge | 0.3085 ± 0.0045 | 0.3904 ± 0.0130 |
> |      | GRAPHSPA | 0.3270 ± 0.0083 | 0.4116 ± 0.0051 |
> | 0.7 | PageRank | 0.3259 ± 0.0034 | 0.4189 ± 0.0112 |
> |      | Degree   | 0.3066 ± 0.0159 | 0.4145 ± 0.0053 |
> |      | DropEdge | 0.3105 ± 0.0116 | 0.3750 ± 0.0051 |
> |      | GRAPHSPA | **0.3326 ± 0.0068** | **0.4245 ± 0.0066** |
> | 0.6 | PageRank | 0.3405 ± 0.0121 | 0.3976 ± 0.0024 |
> |      | Degree   | 0.3256 ± 0.0014 | 0.4189 ± 0.0081 |
> |      | DropEdge | 0.3138 ± 0.0171 | 0.3838 ± 0.0035 |
> |      | GRAPHSPA | **0.3481 ± 0.0053** | **0.4279 ± 0.0245** |
> | 0.5 | PageRank | 0.3456 ± 0.0122 | 0.4079 ± 0.0088 |
> |      | Degree   | 0.3336 ± 0.0156 | 0.4276 ± 0.0092 |
> |      | DropEdge | 0.3248 ± 0.0155 | 0.3706 ± 0.0130 |
> |      | GRAPHSPA | **0.3499 ± 0.0075** | **0.4339 ± 0.0231** |
>
> ---
> We additionally provide the statistics of all newly added datasets used in the extended evaluation.
>
> | Dataset     | #Nodes  | #Edges        | #Features | #Classes |
> |-------------|---------|---------------:|-----------:|---------:|
> | Actor       | 7,600   | 26,752         | 932        | 5        |
> | Chameleon   | 2,277   | 36,101         | 2,325      | 5        |
>
> ---
> #### **References**
> [1] Jin et al., “Graph Structure Learning for Robust Graph Neural Networks,” KDD 2020.
> [2] Liu et al., “PTDNet: Learning to Drop Task-Irrelevant Edges for Robust GNNs,” NeurIPS 2021.
> [3] Zhang et al., “A Survey on Adversarial Attacks and Defenses in Graph Machine Learning,” 2022.
> [4] Chen et al., “Demystifying Graph Sparsification Algorithms in Graph Properties Preservation,” PVLDB 2023.

---

> > ### Author Response · Authors · 2025-11-21
> >
> > > #### **Q2. What is the impact of different noise distributions on stability?**
> >
> > Thank you for raising this question. In the current version of our paper, we evaluate robustness using only random noise injection, following the widely used protocol of Jin et al. (2021). This setting assesses how well the model tolerates unbiased residual noise that remains after sparsification.
> >
> > However, we fully agree with the reviewer that, in order to more thoroughly demonstrate the encoder’s robustness and generalization ability, it is important to evaluate a broader set of structured and adversarial noise distributions such as adversarial noise, targeted noise, poisoning-style perturbations, and feature-dependent noise. These noise types challenge the model in fundamentally different ways and can reveal deeper insights into stability under heterogeneous perturbations.
> >
> > Because our current experiments already span multiple datasets and sparsification baselines, we were not able to finalize an extended noise-distribution study within the original submission timeline. We are now conducting additional experiments using adversarial, targeted, and structured noise distributions, and we plan to update the results during the rebuttal period as soon as they are completed.
> >
> > Thank you for the suggestion. We will incorporate this improvement in the revised version, and we appreciate the reviewer’s feedback, which helps strengthen the completeness of our robustness evaluation.

---

> > > ### Author Response · Authors · 2025-11-30
> > >
> > > > #### **Q2. What is the impact of different noise distributions on stability?**
> > >
> > > We deeply agree that verifying stability under more diverse and challenging noise distributions, beyond simple random noise, is essential to firmly establish the reliability of the proposed framework. Reflecting the reviewer’s suggestion, we have significantly expanded our evaluation in the revised manuscript. We performed a comprehensive assessment by including not only random noise but also homophily-breaking noise, which hinders GNN learning, and adversarial noise, which is designed to intentionally mislead the classifier.
> > >
> > > As demonstrated in Figure 2 of Section 5.2, GRAPHSPA exhibits superior stability and robustness compared to all baseline models across all three noise types. Existing sparsification methods typically rely on preserving specific structural properties. Consequently, they show significant vulnerability when subjected to homophily-breaking or adversarial noise that fundamentally disrupts those very properties. In contrast, GRAPHSPA maintains the highest accuracy consistently across all conditions, showing a much more gradual decline in performance as the noise ratio increases.
> > >
> > > This robustness stems from the design of GRAPHSPA, which maximizes mutual information to preserve intrinsic signals rather than superficial structures, and concurrently employs flatness-aware optimization to prevent the encoder from overfitting to residual noise. These detailed experimental results and analyses have been fully reflected in the revised paper.
> > >
> > > We sincerely thank the reviewer for the insightful question, which has greatly helped in enhancing the completeness of our work.

---

### Author Response · Authors · 2025-12-01

We thank the reviewers for their constructive feedback and have revised the manuscript accordingly. We believe that the updated version substantially improves the clarity, completeness, and technical depth of the work.

Below, we summarize the main changes made in the revised manuscript.

- We added four new sparsification baselines (Spectral, Shortest-Path, Topo-Cycle, Topo-Cluster), and included detailed descriptions of each method in the Baselines subsection of Section 5.  (`NpUb`, `8WsS`)

- We updated Figure 2 with additional experiments evaluating robustness under diverse noise types (random, homophily-breaking, adversarial). (`pq7G`)

- We expanded Table 1 to include experimental results on large-scale datasets, and added further analysis of these results in Section 5.1. (`pq7G`, `PDrR`, `NpUb`)

- We updated Figure 1 and Figure 2 to incorporate the results of these newly added baselines across all benchmark datasets. (`NpUb`, `8WsS`)

- We added heterophilic dataset analysis in Appendix D.2, with corresponding experimental results summarized in the newly added Figure 4. (`pq7G`, `8WsS`)

- We included the time complexity of the newly added baselines in Table 4, and reorganized Appendix C.2 to more clearly explain the associated computational analysis. (`pq7G`, `PDrR`, `NpUb`)

- Given the increased number of baselines, we replaced shaded standard-deviation bands with p-values to present statistical significance more clearly.

- We added an ablation study on negative sampling temperature in Appendix D.3. (`pq7G`)
- We made minor revisions throughout the paper to clarify the contributions, while leaving the methodology unchanged.

We believe these revisions meaningfully strengthen the manuscript and directly address the key concerns raised during the review process. We sincerely appreciate the reviewers’ careful evaluation and hope that the improvements are reflected in their final assessment.

---

### Author Response · Authors · 2025-12-01

We sincerely thank the ICLR community for their time and constructive engagement, and we conclude with a summary of the reviews and discussions.

---

> ### **Strengths**
- **Robust constrained optimization framework ensuring stable** (`pq7G`, `PDrR`)
    - The augmented Lagrangian with progressive sparsification provides a consistently stable optimization process.
-  **Insightful, principled, and novel integration of core components** (`pq7G`, `PDrR`, `NpUb`)
    - The framework combines MI-based self-supervision, augmented Lagrangian optimization, and flatness-aware training into a coherent and novel design.
- **Practical contribution addressing both label dependency and residual noise** (`PDrR`, `NpUb`)
    - The method simultaneously tackles two central issues: removing label dependency and mitigating residual noisy edges, filling an important practical gap in existing sparsification approaches.
-  **Clear presentation and high readability** (`8WsS`)
    - The paper presents key ideas and technical details clearly, making the overall structure easy to understand.

> ### **Clarification for Concerns**
- **Scalability and computational overhead** (`pq7G`, `PDrR`, `NpUb`)
    - Additional experiments on large-scale datasets demonstrate sufficient scalability, and we show that the proposed method is more efficient in time complexity compared to spectral, path-based, and topology-based sparsifiers.
- **Limited baseline diversity** (`NpUb`, `8WsS`)
    - We newly implemented four sparsifiers and substantially expanded the comparison suite, including reproducing methods without official implementations.
- **Dataset diversity** (`pq7G`, `PDrR`, `8WsS`)
    - We added heterophilous datasets and large-scale datasets, significantly broadening the structural and scale diversity of the evaluation.
- **Suitability of the MI objective** (`PDrR`)
    - We examined alternative objectives and explained that, in large-scale and label-free settings, MI objective provides the most stable and unbiased learning signal.
- **Lack of ablation study** (`pq7G`)
    - We conducted an additional ablation study that examined both the negative sampling temperature and the robustness under diverse noise types.

We appreciate the constructive feedback, which has helped improve the clarity and technical foundation of our work.

Best regards,
Authors

---

### Note · Authors · 2026-01-13

I have read and agree with the venue's withdrawal policy on behalf of myself and my co-authors.